# Probing PAC1 receptor activation across species with an engineered sensor

Reto B Cola[1], Salome N Niethammer[2], Preethi Rajamannar[3], Andrea Gresch[1], Musadiq A Bhat[1], Kevin Assoumou[4], Elyse T Williams[5], Patrick Hauck[5], Nina Hartrampf[5], Dietmar Benke[1,6], Miriam Stoeber[4], Gil Levkowitz[3], Sarah Melzer[2], Tommaso Patriarchi[1,6]*

[1]Institute of Pharmacology and Toxicology, University of Zürich, Zurich, Switzerland; [2]Medical University of Vienna, Center for Brain Research, Department for Neuronal Cell Biology, Vienna, Austria; [3]Department of Molecular Neuroscience & Department of Molecular Cell Biology, Weizmann Institute of Science, Rehovot, Israel; [4]Department of Cell Physiology and Metabolism, University of Geneva, Geneva, Switzerland; [5]Department of Chemistry, University of Zürich, Zürich, Switzerland; [6]Neuroscience Center Zurich, University and ETH Zürich, Zürich, Switzerland

**Abstract** Class-B1 G-protein-coupled receptors (GPCRs) are an important family of clinically relevant drug targets that remain difficult to investigate via high-throughput screening and in animal models. Here, we engineered PAClight1$_{P78A}$, a novel genetically encoded sensor based on a class-B1 GPCR (the human PAC1 receptor, hmPAC1R) endowed with high dynamic range ($\Delta F/F_0$ = 1100%), excellent ligand selectivity, and rapid activation kinetics ($\tau_{ON}$ = 1.15 s). To showcase the utility of this tool for in vitro applications, we thoroughly characterized and compared its expression, brightness and performance between PAClight1$_{P78A}$-transfected and stably expressing cells. Demonstrating its use in animal models, we show robust expression and fluorescence responses upon exogenous ligand application ex vivo and in vivo in mice, as well as in living zebrafish larvae. Thus, the new GPCR-based sensor can be used for a wide range of applications across the life sciences empowering both basic research and drug development efforts.

*For correspondence: patriarchi@pharma.uzh.ch

## eLife assessment

This **fundamental** paper reports a new biosensor to study G protein-coupled receptor activation by the pituitary adenylyl cyclase-activating polypeptide (PACAP) in cell culture, ex vivo (mouse brain slices), and in vivo (zebrafish, mouse). **Convincing** data are presented that show the new sensor works with high affinity in vitro, while requiring very high (non-physiological) concentrations of exogenous PACAP when applied to intact tissues. The sensor has not yet been used to detect endogenously released PACAP, raising questions about whether the sensor can be used for its intended purpose. While further work must be pursued to achieve broad in vivo applications under physiological conditions, the new tool will be of interest to cell biologists, especially those studying the large and significant GPCR family.

## Introduction

Class-B1 G-protein-coupled receptors (GPCRs) represent an important sub-group of peptide-sensing GPCRs, that are the focus of intense and rapidly expanding drug development efforts (*Hauser et al., 2017*), driven by extremely successful examples of peptide agonists used in the clinical treatment of metabolic human diseases, such as type-2 diabetes and obesity (*Wang et al., 2022b*). One such

peptide–GPCR system that has shown growing potential for targetability in the treatment of human disorders is the Pituitary Adenylate Cyclase Activating Peptide (ADCYAP1 or PACAP) and its receptors. PACAP is an endogenous 38-amino acid peptide that is among the most phylogenetically conserved peptides (*Johnson et al., 2020*). Its shorter C-terminally truncated form (i.e., PACAP$_{1-27}$) has 68% homology with the vasoactive intestinal peptide (VIP) (*Dickson and Finlayson, 2009*). In fact, VIP and PACAP share a subfamily of class-B1 GPCRs, of which the two receptors VPAC1 and VPAC2 can both be equipotently activated by VIP and PACAP (*Dickson and Finlayson, 2009*). The third receptor in this subfamily, that is the PAC1 receptor (PAC1R, also known as ADCYAP1R1), however, has a reported affinity for PACAP that is 100- to 1000-fold higher than its affinity to VIP (*Dickson and Finlayson, 2009*). The tissue distribution of PACAP and its receptors is widespread and they can be found throughout the central and peripheral nervous system, the immune system, in endocrine glands as well as in other organ systems and in many cancerous tissues (*Dickson and Finlayson, 2009*; *Hashimoto et al., 1996*; *Condro et al., 2016*; *PACAP, 2016*; *Blechman and Levkowitz, 2013*). A large body of evidence has linked the PACAP/PAC1R system to protective functions in the nervous and immune systems, as well as to stress- and anxiety-related behaviors (in particular post-traumatic stress disorder), migraine, nociception, thermoregulation, sleep/wake cycles, and reproductive functions (*Dickson and Finlayson, 2009*; *PACAP, 2016*; *Biran et al., 2020*), making it a peptide signaling system of high clinical relevance.

Recently, genetically encoded GPCR-based sensors have been developed that enable the direct optical detection of GPCR activation by agonist ligands with high sensitivity and spatiotemporal resolution (*Patriarchi et al., 2018*; *Patriarchi et al., 2020*; *Patriarchi et al., 2019*; *Sun et al., 2018*; *Peng et al., 2020*; *Oe et al., 2020*; *Duffet et al., 2022*; *Duffet et al., 2023*; *Kagiampaki et al., 2023*). While these tools hold great potential for drug development, pharmacology, and neuroscience applications, only a few of them are built from class-B1 peptide-sensing GPCRs (*Duffet et al., 2023*; *Wang et al., 2022a*). Thus, the development of new biosensors based on peptide-sensing GPCRs would greatly benefit the community and serve as a powerful resource for drug screening and life sciences.

In this work, we engineered PAClight1$_{P78A}$, an ultrasensitive indicator based on the human PAC1 receptor (hmPAC1R). To establish this as a tool for peptide drug screening, we thoroughly characterized its dynamic range, as well as its optical, kinetic, signaling, and pharmacological properties in vitro. We further generated stable cell lines and compared them to transfected cells for ligand-induced fluorescence responses using flow cytometry. Additionally, we tested the potential of PAClight1$_{P78A}$ as a tool to study ligand binding and diffusion in animal models. To this end, we examined the sensitivity and specificity of PAClight1$_{P78A}$ in acute mouse brain slices in response to application of PACAP$_{1-38}$. Moreover, we verified ligand detection in vivo using fiber photometry recording and intracerebral microinfusions in behaving mice.

Finally, we demonstrate that the sensor expresses well and produces a large fluorescent response to PACAP$_{1-38}$ following microinjection in the brain of living zebrafish larvae, opening new opportunities for the development and testing of drugs targeting this receptor in the central nervous system.

Overall, our new sensor expands the class of genetically encoded optical tools that can be used for GPCR-targeted HTS assays, reducing the demands on time and costly reagents, and provides new opportunities for functionally testing drugs that target the PAC1R pathway directly in living animals.

## Results
### Development of an ultrasensitive PAC1R-based sensor

To develop a highly sensitive indicator of hmPAC1R activation, we followed a protein engineering approach that we and others recently established (*Patriarchi et al., 2018*; *Sun et al., 2018*; *Duffet et al., 2022*). We used the human PAC1Rnull splice isoform as a protein scaffold (hmPAC1R) and constructed an initial sensor prototype in which we replaced the entire third intracellular loop (ICL3, residues Q336–G342) with a module containing circularly permuted green fluorescent protein (cpGFP) from dLight1.3b (*Patriarchi et al., 2018*; *Figure 1—figure supplement 1a*). This initial prototype construct, named PAClight0.1, was expressed on the surface of HEK293T cells, but showed clear intracellular retention and a very weak average fluorescent response to bath application of PACAP$_{1-38}$ ($\Delta F/F_0$ = 43.4%, *Figure 1—figure supplement 1b*). Using this sensor as template, we conducted a small-scale screen to identify the optimal insertion site by reintroducing amino acids from the original

ICL3 of hmPAC1R on both sides of the cpGFP module. This led us to the identification of a second mutant, in which Q336 was reintroduced before the cpGFP module, that showed improved average fluorescent response to the ligand ($\Delta F/F_0$ = 343%, *Figure 1—figure supplement 1c*). To improve membrane expression of the PAClight mutants, we next investigated the effect of point mutations on their C-terminus, via site-directed mutagenesis. Mutation into alanine of residue T468, a reported post-translational modification site (https://www.phosphosite.org) on the C-terminus of the receptor (*Hornbeck et al., 2004*), improved the overall surface expression of the sensor in HEK293T cells and further increased the observed average fluorescent response ($\Delta F/F_0$ = 516%, *Figure 1—figure supplement 1d–g*). To further improve the fluorescent response of PAClight, we screened a subsequent library of sensor variants with mutations targeted to the second intracellular loop (ICL2). A similar approach was previously shown to boost the fluorescent response of other GPCR-based sensors, for example dLight1.3b or OxLight1 (*Patriarchi et al., 2018*; *Duffet et al., 2022*). Our functional screen of a library of ICL2 mutants led to the identification of one variant comprising three point mutations (F259K/F260K/P261K) that displayed a largely improved fluorescent response to PACAP$_{1-38}$ ($\Delta F/F_0$ = 942%, *Figure 1—figure supplement 1d–g*). Upon introduction of the C-term T468A mutation on the ICL2 F259K/F260K/P261K background, we obtained an improved version of the sensor, named PAClight1, with excellent surface expression and fluorescent response ($\Delta F/F_0$ = 1037%, *Figure 1—figure supplement 1g*). Given the high degree of sequence and structural similarity, as well as the broad activation of endogenous VPAC(VIP and PACAP receptors) receptors by VIP and PACAP (*Kobayashi et al., 2020*), we next asked whether the hmPAC1R-based sensor would also respond to VIP. Indeed, application of a high concentration of VIP onto sensor-expressing cells caused a large fluorescent response, corresponding to more than half of the response to PACAP$_{1-38}$ ($\Delta F/F_0$ = 654%, *Figure 1—figure supplement 2a, b*). In an effort to eliminate the response to VIP and obtain a PACAP-specific sensor, we screened a small library of sensors containing single-point mutations into alanine that were inspired by simulations on the binding free energy of PACAP and VIP on the PAC1R (*Liao et al., 2021*). Furthermore, at this stage we included the naturally occurring splice mutant PAC1R 'short' (reported to have lower affinity toward VIP; *Blechman and Levkowitz, 2013*) into the extracellular ligand-binding domain (ECD) of the sensor, as well as two structure-guided point mutants. Through this screening, we identified a single mutation (P78A) that abolished the sensor response to VIP while leaving unaltered the response to PACAP$_{1-38}$ (*Figure 1—figure supplement 2c, d*). The final sensor construct, named PAClight1$_{P78A}$ (*Figure 1a, b*, *Supplementary file 1*), includes all the above-mentioned mutations and combines a very large dynamic range with excellent PACAP selectivity and good membrane expression, and was thus selected for further in vitro characterization.

## In vitro characterization of PAClight1$_{P78A}$

The newly developed PAClight1$_{P78A}$ sensor displays an average fluorescent response of 1066% $\Delta F/F_0$ in HEK293T cells ($n$ = 3, five ROIs each) to bath application of 10 µM PACAP$_{1-38}$ (*Figure 1b–d*). We verified that sensor expression and function are not drastically affected by the cell type in which it is expressed by testing it in primary cultured neurons. Neurons were virally transduced using an adeno-associated virus (AAV) for expressing the PAClight1$_{P78A}$ sensor under control of a human synapsin-1 promoter. Two to three weeks after transduction we verified excellent expression of the probe on the plasma membrane of the neurons with no appreciable intracellular retention. Under these conditions, the sensor showed a fluorescence response of 883% $\Delta F/F_0$ upon bath application of the PACAP$_{1-38}$ ligand (*Figure 1c, d*).

We next set out to determine the sensor's excitation and emission spectra in vitro in HEK29T cells (*Figure 1e*). The excitation maxima in the absence and presence of 10 µM PACAP$_{1-38}$ were identified at 504 and 498 nm, respectively. The isosbestic point at which the excitation is independent of the absence or presence of PACAP$_{1-38}$ is located at 420 nm. The maxima for the emission spectra in the absence of PACAP$_{1-38}$ was identified at 520 nm and in the presence of PACAP$_{1-38}$ at 514 nm.

Our previous work on the development of another class-B1 sensor based on the GLP1 receptor, led us to discover that the kinetics of the sensor's response can be used to infer the occupancy of the ECD by an antagonist peptide (*Duffet et al., 2023*). Given that all class-B1 GPCRs share a similar ECD high-affinity ligand-binding mechanism, we performed similar experiments to determine whether PAClight1$_{P78A}$ could also be used in a similar manner. To do so, we monitored the sensor's response during application of PACAP$_{1-38}$ alone or in the presence of PACAP$_{6-38}$, an antagonist peptide, in the

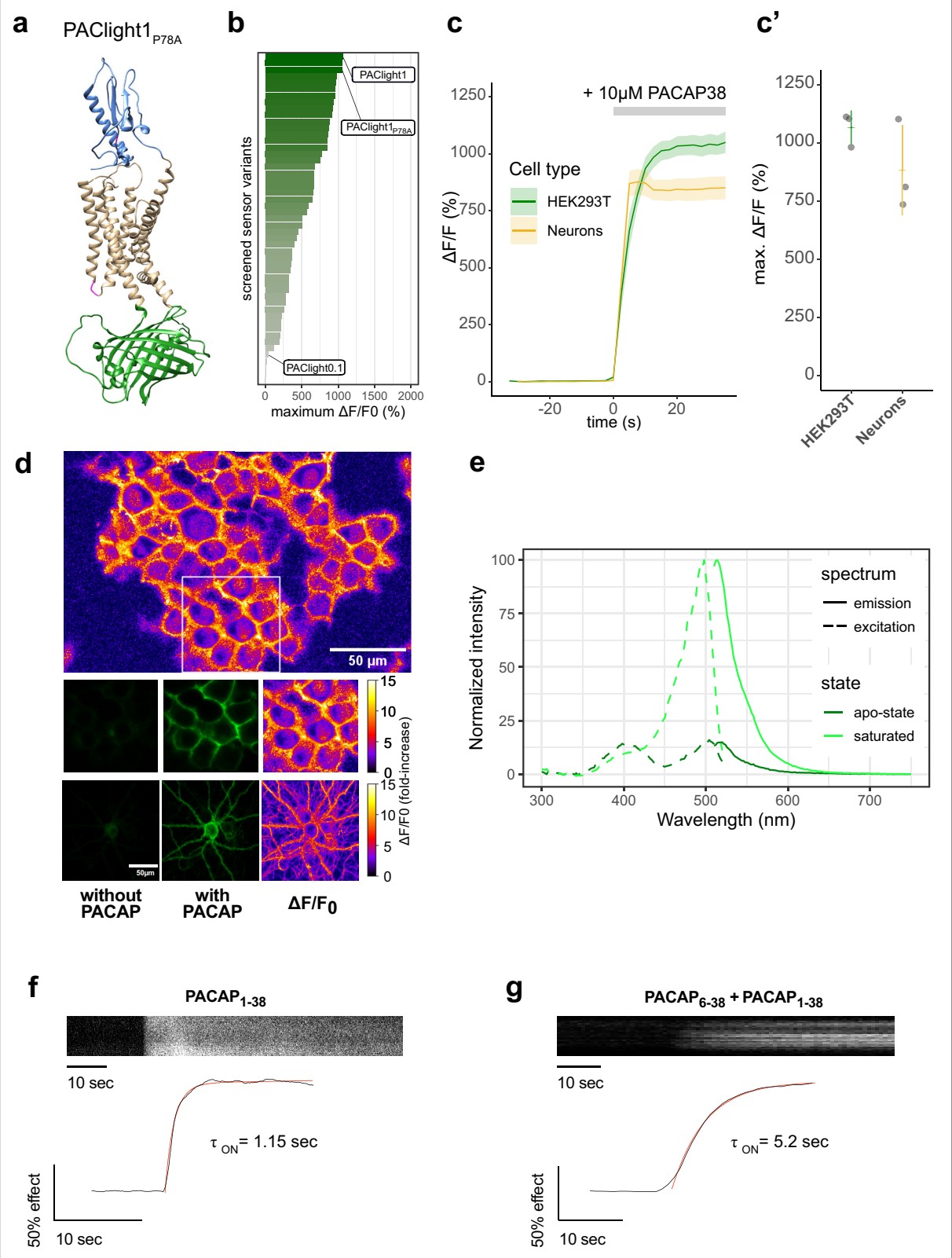

**Figure 1.** Design and in vitro optical properties of PAClight1P78A. (**a**) Protein structure of PAClight1P78A as predicted by AlphaFold2 (*Jumper et al., 2021*). The transmembrane and intracellular domain of the PAC1Rnull backbone is depicted in beige. The extracellular domain, which is crucial for ligand specificity and affinity, is colored in blue. The circularly permuted green fluorescent protein (cpGFP) module replacing the third intracellular loop is colored in green. Point-mutations inserted in the second intracellular loop as well as on the extracellular domain are depicted in magenta. (**b**) Bar

*Figure 1 continued on next page*

*Figure 1 continued*

chart depicting the maximum dynamic range obtained for all tested mutants with acceptable expression on the plasma membrane. Bars are ordered by dynamic range and color coded by the average of the maximum dynamic range recorded. The bars representing the prototype variant PAClight_V0.1, PAClight1$_{P78A}$, and PAClight1 are indicated. (**c**) Line plots depicting the maximum activation (mean ± standard error of the mean [SEM]) of PAClight1$_{P78A}$ expressed in HEK293T cells and rat primary neurons. Upon bath application of 10 µM PACAP$_{1-38}$, PAClight1$_{P78A}$ reaches a mean $\Delta F/F_0$ of 1066% in HEK293T cells ($n = 3$, five regions of interest [ROIs]) and a mean $\Delta F/F_0$ of 883% in rat primary neurons ($n = 3$, five ROIs). (**c'**) Scatter plot representation of the maximum $\Delta F/F_0$ of the individual replicates shown in c. Confocal image acquisition was performed at a frame rate of 1 frame/2.53 s (~0.4 Hz). (**d**) Representative examples of the expression of PAClight1$_{P78A}$ in HEK293T cells (low magnification overview and middle row of images) and rat primary neurons (bottom row) with pixel-wise quantification and depiction of the dynamic range in $\Delta F/F_0$ upon bath application of 10 µM PACAP$_{1-38}$. Top: Overview micrograph of HEK293T cells color coded in pixel-wise $\Delta F/F_0$. The white square represents the ROI represented below in the middle row of images. Middle: Selected ROI before (left) and after (middle and right) peptide application. Bottom: Rat primary neuron-expressing PAClight1$_{P78A}$ before (left) and after (middle and right) peptide application. The color bars represent the look-up table used for visualization of the pixel-wise $\Delta F/F_0$. (**e**) One-photon excitation (dashed lines) and emission (solid lines) spectra of PAClight1$_{P78A}$ in the absence (apo-state, dark green) and the presence (saturated state, light green) of 10 µM PACAP$_{1-38}$. The excitation maximum of the saturated state is at 498 nm. The isosbestic point is at 420 nm. Emission maximum in of the saturated state is at 514 nm. $n = 4$ (2 replicates each measured on 2 independent days). (**f**) Activation kinetics of PAClight1$_{P78A}$ upon application of PACAP$_{1-38}$ (10 µM) measured via time-lapse imaging. A representative kymograph of sensor fluorescence on the surface of a HEK293 cell is shown on top. The normalized fluorescence response trace is shown at bottom along with the calculated one-phase association curve fit and activation time constant. The trace shown is the average of three independent experiments. (**g**) Same as (**f**) but in the presence of PACAP$_{6-38}$ (10 µM) in the bath.

The online version of this article includes the following source data and figure supplement(s) for figure 1:

**Source data 1.** Source data for graphs and bar plots in *Figure 1*.

**Figure supplement 1.** Optimizing the fluorescent response of PAClight sensors.

**Figure supplement 2.** Engineering PACAP selectivity in PAClight1$_{P78A}$.

buffer surrounding the cells. We then determined the activation time constant of the fluorescent response in both conditions. The PAClight1$_{P78A}$ response was strikingly slower (approximately fourfold) in the presence of the extracellular antagonist, and was in the range of 1 s in its absence (*Figure 1f, g*). Thus, the kinetics of PACligth1$_{P78A}$ response could be used to investigate or screen for factors that influence the ECD–PACAP interaction and potentially the speed of signal transduction through conformational activation of the receptor.

To investigate the stability of the fluorescent response to bath application of PACAP$_{1-38}$, a long-term imaging experiment was performed at room temperature with 1 frame (1024 × 1024 pixels, 2× line-averaging) acquired every minute over the time course of 150 min (*Figure 2a*). After bath application of 200 nM PACAP$_{1-38}$ (slow diffusion of the peptide with 10× dilution from 2 µM to 200 nM) there was no decrease in signal, nor any internalization observed. After >100 frames of acquisition, we applied a saturating bolus of the peptide PAC1R antagonist Max.d.4. Within 45–50 frames after application of the PAC1R antagonist, the signal intensity steadily decreased and started to plateau slightly above baseline levels. This indicates, that Max.d.4 can outcompete PACAP$_{1-38}$ at PACligth1$_{P78A}$, albeit not to a full extent under the specified experimental conditions.

Next, we screened a range of different peptides at saturating concentrations (10 µM) for potential activation of the PAClight1$_{P78A}$ sensor (*Figure 2b*). Mammalian PACAP$_{1-38}$ displayed the strongest potency in PAClight1$_{P78A}$ activation (1052% $\Delta F/F_0$, $t_{(3)} = 40.31$, 95% CI [9.69, 11.35], p < 0.001). Because the amino acid sequence of PACAP is rather well conserved throughout phylogeny, we also tested chicken PACAP$_{1-38}$ (chPACAP38), as well as the zebrafish PACAP2$_{1-27}$ (zfPACAP2) for potency on the PAClight1$_{P78A}$ sensor. Both of these homologs of the mammalian PACAP$_{1-38}$ activated the PAClight1$_{P78A}$ sensor with strong but slightly reduced potency (chPACAP38: 896% $\Delta F/F_0$, $t_{(4)} = 11.17$, 95% CI [6.73, 11.19], p = 0.0066; zfPACAP2: 898% $\Delta F/F_0$, $t_{(3)} = 11.49$, 95% CI [6.46, 11.41], p = 0.024). Furthermore, we also screened the sand fly salivary gland-derived peptide Maxadilan, which was previously identified to be a specific ligand of the PAC1R but not to the VPAC1 and VPAC2 receptors (*Lerner et al., 2007*). Consistent with the reported activity of Maxadilan on the hmPAC1Rnull receptor, we detected activation of PAClight1$_{P78A}$ by Maxadilan, however, with lower potency than mammalian PACAP$_{1-38}$ (416% $\Delta F/F_0$, $t_{(3)} = 9.47$, 95% CI [2.76, 5.56], p = 0.04). The reduced potency of Maxadilan for PAClight1$_{P78A}$ might be a consequence of the ligand specificity determining point mutation (P78A) introduced into the extracellular domain of the PAClight1$_{P78A}$ sensor. The *Drosophila* gene *amn* (amnesiac) was previously shown to be homologous to the mammalian gene encoding PACAP (*Adcyap1*) and *amn Drosophila* mutants show memory impairments similar to rodent PACAP/PAC1R

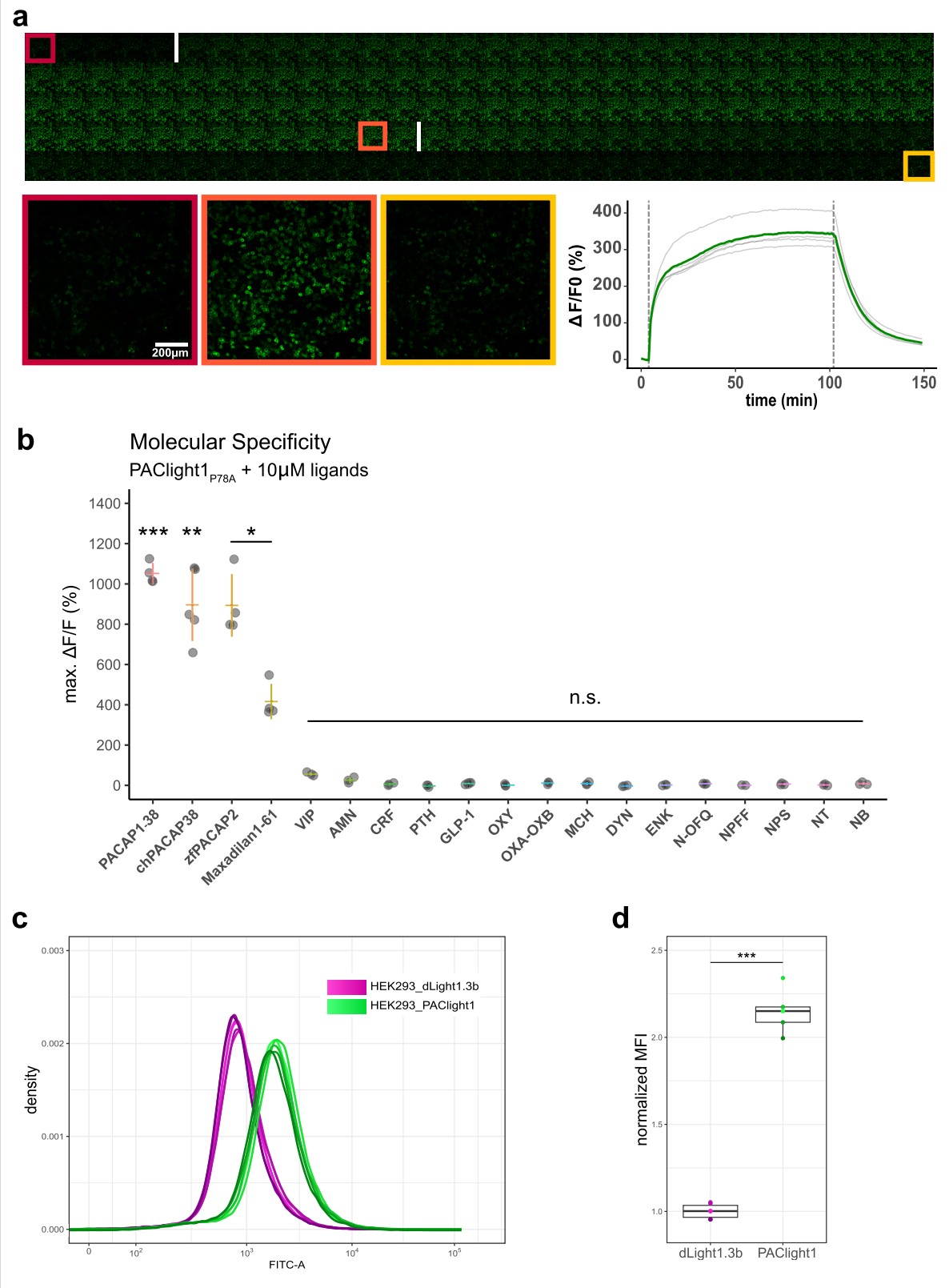

**Figure 2.** Pharmacological characterization of PAClight1$_{P78A}$. (**a**) PAClight1$_{P78A}$ fluorescent response to 200 nM PACAP$_{1-38}$ bath application over extended periods at room temperature. No internalization of the sensor expressed on the plasma membrane is observed throughout the full time course of >90 min. The fluorescent response of PAClight1$_{P78A}$ to PACAP$_{1-38}$ can be reversed with competitive binding of the peptidergic PAC1R antagonist Max.d.4 (a Maxadilan derivative). Please note that the seemingly slow activation rate of PAClight1$_{P78A}$ in this experiment is due to the experimental setup

*Figure 2 continued*

(see Methods section) and slow diffusion of PACAP$_{1-38}$ throughout the well. Top: Downscaled gallery view of the acquired time series (total of 150 frames; 1 frame/min) from top-left to bottom-right. Application time points of PACAP$_{1-38}$ (after frame Nr. 5) and Max.d.4 (after frame Nr. 103) are depicted as white vertical lines between the frames. The red, orange, and yellow rectangles indicate the frames used for representative higher magnification inserts shown below (bottom left). Bottom right: Line plot depicting the time course of the PAClight1$_{P78A}$ response across five rectangular regions of interest (ROIs) distributed across the whole field of view. (**b**) PAClight1$_{P78A}$ is highly specific to PACAP and does not respond to vasoactive intestinal peptide (VIP). PAClight1$_{P78A}$ can be activated by PACAP homologs found in the chicken (gallus gallus, chPACAP38) as well as in the zebrafish (*Danio rerio*, zfPACAP2). PAClight1$_{P78A}$ is further partially activated by the PAC1R-specific ligand Maxadilan$_{1-61}$, which is expressed endogenously in the sand fly (Lutzomyia longipalpis) salivary gland. None of the other tested class-A and -B1 G-protein-coupled receptor (GPCR) ligands was found to activate PAClight1$_{P78A}$. Abbreviations: AMN: amnesiac, CRF: corticotropin-releasing factor, PTH: parathyroid hormone, GLP-1: glucagon-like peptide, OXY: oxytocin, OXA–OXB: orexin-A and -B, MCH: melanin-concentrating hormone, DYN: dynorphin, ENK: enkephalin, N-OFQ: nociception, NPFF: neuropeptide FF, NPS: neuropeptide S, NT: neurotensin, NB: neuromedin B. Single datapoints represent one replicate average obtained from five ROIs per replicate. The extent of the colored vertical bar represents 1 standard deviation. The *y*-axis location of the colored horizontal bar indicates the average across all replicates. Number of replicates per ligand: $n$ = 5 for chPACAP38 and GLP-1; $n$ = 4 for PACAP$_{1-38}$, zfPACAP2, and Maxadilan$_{1-61}$; $n$ = 3 for VIP, AMN, CRF, PTH, OXY, OXA–OXB, MCH, DYN, ENK, N-OFQ, NPS, NT, and NB; $n$ = 2 for NPFF. Asterisks represent statistical significance of Hochberg-corrected p values of multiple one-sample *t* tests. (**c**) PAClight1 has significantly brighter baseline fluorescence than the dopamine sensor dLight1.3b. The peak of the dLight1.3b FITC-A density curve coincides with the start of the uphill slope of the PAClight1 FITC-A density curve. Note the bi-exponential scaling of the *x*-axis. $n$ = 6 for dLight1.3b, $n$ = 5 for PAClight1. (**d**) Quantification of the average of the median fluorescence intensity (MFI) across all replicates and normalization to the group MFI of dLight1.3b show a 2.15-fold increased basal brightness of PAClight1 over dLight1.3b ($t_{(4.74)}$ = −19.254, p < 0.0001, 95% CI [−1.30,−0.99], two-sided two-sample Welch's *t* test).

The online version of this article includes the following source data and figure supplement(s) for figure 2:

**Source data 1.** Source data for graphs and bar plots in *Figure 2*.

**Figure supplement 1.** Characterization of PAClight1$_{P78A}$ coupling to transducer proteins NanoLuc complementation assays were employed to measure the ability of human SmBiT-PAC1R or SmBiT-PAClight1$_{P78A}$ to recruit LgBiT-miniGs (**a**), -miniGsq (**b**), or -β-arrestin2 (**c**) in an agonist-induced manner.

**Figure supplement 2.** Development of a PAClight1$_{P78A}$-ctrl sensor.

mutants (*Feany and Quinn, 1995*). We therefore also tested whether this insect peptide encoded by *amn* would activate PAClight1$_{P78A}$, but no significant activation above baseline was observed (26% $\Delta F/F_0$, $t_{(2)}$ = 3.2, 95% CI [−0.09, 0.61], p = 0.69). As shown above (*Figure 1—figure supplement 2*), PAClight1$_{P78A}$ was specifically optimized for VIP non-responsive properties. Therefore, the response of PAClight1$_{P78A}$ to VIP in this specificity screen was also not significantly above baseline (57% $\Delta F/F_0$, $t_{(2)}$ = 11.09, 95% CI [0.35, 0.79], p = 0.12). Other peptides tested in this screen included other class-B1 GPCR ligands (corticotropin-releasing factor (CRF), parathyroid hormone (PTH), and glucagon-like peptide 1 (GLP-1)), as well as some peptidergic class-A GPCR ligands (oxytocin (OXY), orexin-A and -B (OXA–OXB), melanin-concentrating hormone (MCH), dynorphin (DYN), enkephalin (ENK), nociception (N-OFQ), neuropeptide FF (NPFF), neuropeptide S (NPS), neurotensin (NT), and neuromedin B (NB)). None of these neuropeptides activated the PAClight1$_{P78A}$ sensor above baseline level (*Figure 2b*). Taken together, these results highlight the broad potential applicability of the PAClight1$_{P78A}$ sensor for use in model systems across the phylogenetic tree, as well as its high selectivity for PACAP ligands over VIP and other peptide GPCR ligands.

In order not to induce artificial PACAP signaling and potentially interfere with downstream read-outs when using PAClight1$_{P78A}$, it is important to verify that the sensor does not recruit G proteins and/or β-arrestin. To monitor the capacity of PAClight1$_{P78A}$ to engage these intracellular signaling partners, we performed split NanoLuc complementation assays as in our previous work (*Duffet et al., 2022*; *Kagiampaki et al., 2023*), using either PAClight1$_{P78A}$-SmBiT or, as a positive control, PAC1R-SmBiT fusion constructs, together with LgBiT-miniGs (*Figure 2—figure supplement 1a*), LgBiT-miniGsq (*Figure 2—figure supplement 1b*), or LgBiT-β-arrestin2 (*Figure 2—figure supplement 1c*). As expected, we observed significant miniGs, miniGsq, and β-arrestin2 recruitment to the wild-type PAC1R upon activation with 1 µM of PACAP$_{1-38}$. Yet, we did not detect recruitment of either miniGs, miniGsq, or β-arrestin2 in PAClight1$_{P78A}$-expressing cells upon stimulation with PACAP$_{1-38}$. These results indicate that expression of PAClight1$_{P78A}$ does not artificially induce PACAP-mediated intracellular signaling and is not likely to interfere with endogenous signaling pathways.

## Development of non-responsive PAClight1 control sensors

When employing GPCR sensors in intact living tissue (e.g., when used in animal models) it is often desirable to make use of an appropriate control sensor, in which ligand binding is abolished by virtue

of one or more point mutations in the GPCR-binding pocket. To engineer such control sensors for our PAClight1$_{P78A}$ and PAClight1 sensors, we targeted key residues in the hmPAC1R that interact with residue D3 in PACAP. The carboxylic group of D3 was shown to be crucial for binding affinity and biological activity of PACAP to all three receptor types (i.e., PAC1, VPAC1, and VPAC2) (**Doan et al., 2011**; **Bourgault et al., 2009**). Furthermore, the cryo-EM structure of the hmPAC1R in interaction with PACAP identified residues Y161 and R199 of the hmPAC1R to interact with residue D3 of PACAP (**Kobayashi et al., 2020**). Residue Y161 forms a hydrogen bond with D3, while R199 forms an electrostatic interaction with D3 (**Figure 2—figure supplement 2a, b**). To abolish these interactions, we first mutated R199 into alanine (R199A) on the backbone of PAClight1. This single-point mutant showed drastically reduced average fluorescent response (PAClight1: 1075% $\Delta F/F_0$, PAClight1_R199A: 41.9% $\Delta F/F_0$, **Figure 2—figure supplement 2c**) to 10 µM PACAP$_{1-38}$. To further abolish the remaining response, we additionally mutated Y161 into alanine (Y161A) on the previous backbone (i.e., PAClight1_R199A_Y161A) and named the construct 'PAClight1-ctrl'. This completely reduced the fluorescent response to PACAP$_{1-38}$ to baseline levels (PAClight1-ctrl: 2.87% $\Delta F/F_0$). Next, we cloned these two point mutations into the backbone of PAClight1$_{P78A}$ and observed an equally abolished fluorescent response to PACAP$_{1-38}$ (PAClight1$_{P78A}$: 1034% $\Delta F/F_0$, PAClight1$_{P78A}$-ctrl: 10.44% $\Delta F/F_0$, $t_{(4.03)}$ = 37.89, p < 0.0001, 95% CI [1098.63, 949.02], two-sided two-sample Welch's $t$ test, **Figure 2—figure supplement 2d**) in transfected HEK293T cells, as well as in virally transduced rat primary neuron cultures (**Figure 2—figure supplement 2e**). No drastic differences in basal brightness were observed between PAClight1$_{P78A}$ and PAClight1$_{P78A}$-ctrl constructs (**Figure 2—figure supplement 2f**). In summary, we have developed double point-mutant sensor constructs for both PAClight1$_{P78A}$ and PAClight1 with fully abolished responses to bath application of saturating concentrations of PACAP$_{1-38}$.

## Comparison between transient and stable expression of the sensor

During the development and validation of PAClight1$_{P78A}$, we noticed higher basal brightness levels of PAClight1$_{P78A}$ compared to other GPCR-based fluorescent sensors. To obtain a quantitative comparison, we decided to produce a stable T-Rex HEK293 cell line for inducible expression of PAClight1 (mutant missing P78A ECD mutation). We then used flow cytometry to record multiple replicates of 100,000 cells, and compared their fluorescence intensity to that of a similarly generated cell line expressing the indicator dLight1.3b, which we previously described (**Klein Herenbrink et al., 2022**; **Figure 2c, d**). Visualization of the fluorescent readouts clearly showed a shift of the density curves toward higher fluorescence for HEK293_PAClight1 cells compared to the HEK293_dLight1.3b. Statistical comparison of the distributions of the two cell lines shows 2.15-fold higher basal brightness of PAClight1 over dLight1.3b ($t_{(4.74)}$ = 19.254, p < 0.0001, 95% CI [−1.30,−0.99], two-sided two-sample Welch's $t$ test, **Figure 2d**).

Among the most important potential advantages of using stable cell lines expressing GPCR sensors for pharmacological assays are the homogeneity of expression, reproducibility, and ease of use. To verify that this is indeed the case, we analyzed PAClight1$_{P78A}$-expressing stable cells and compared them to transfected cells using flow cytometry. While a large proportion of cells from the transfected condition expressed the sensor at very low levels or not at all (i.e., were left shifted in the density plot), we also observed a significant proportion of very bright and strongly expressing cells from within the same condition. In fact, there is higher abundance of very bright and strongly expressing cells in the transfected condition than in the induced stable cell line condition (**Figure 3a**). The combination of the higher abundance of very low expression levels and very high expression levels within the transfected condition leads to a significantly increased variability of observed expression levels compared to the induced stable cell line condition. The median of the standard deviation in the FITC-A channel across the titration series was 62.6% smaller in the stable PAClight1$_{P78A}$ relative to the transfected PAClight1$_{P78A}$ condition (p = 0.0142, **Figure 3b**). The variability of cells transfected with PAClight1 was very comparable and was only 8.6% lower than in the PAClight1$_{P78A}$-transfected condition (p = 0.5966, **Figure 3b**). The difference of 8.6% (even though statistically unsignificant) might partially be explained by slightly different performance of PAClight1$_{P78A}$ vs. PAClight1 in dynamic range. p values represent the result of Dunnett's test for correction of multiple comparison that was performed after the statistically significant results of an omnibus analysis of variance (ANOVA; $F_{(2,30)}$ = 4.321, p = 0.0224).

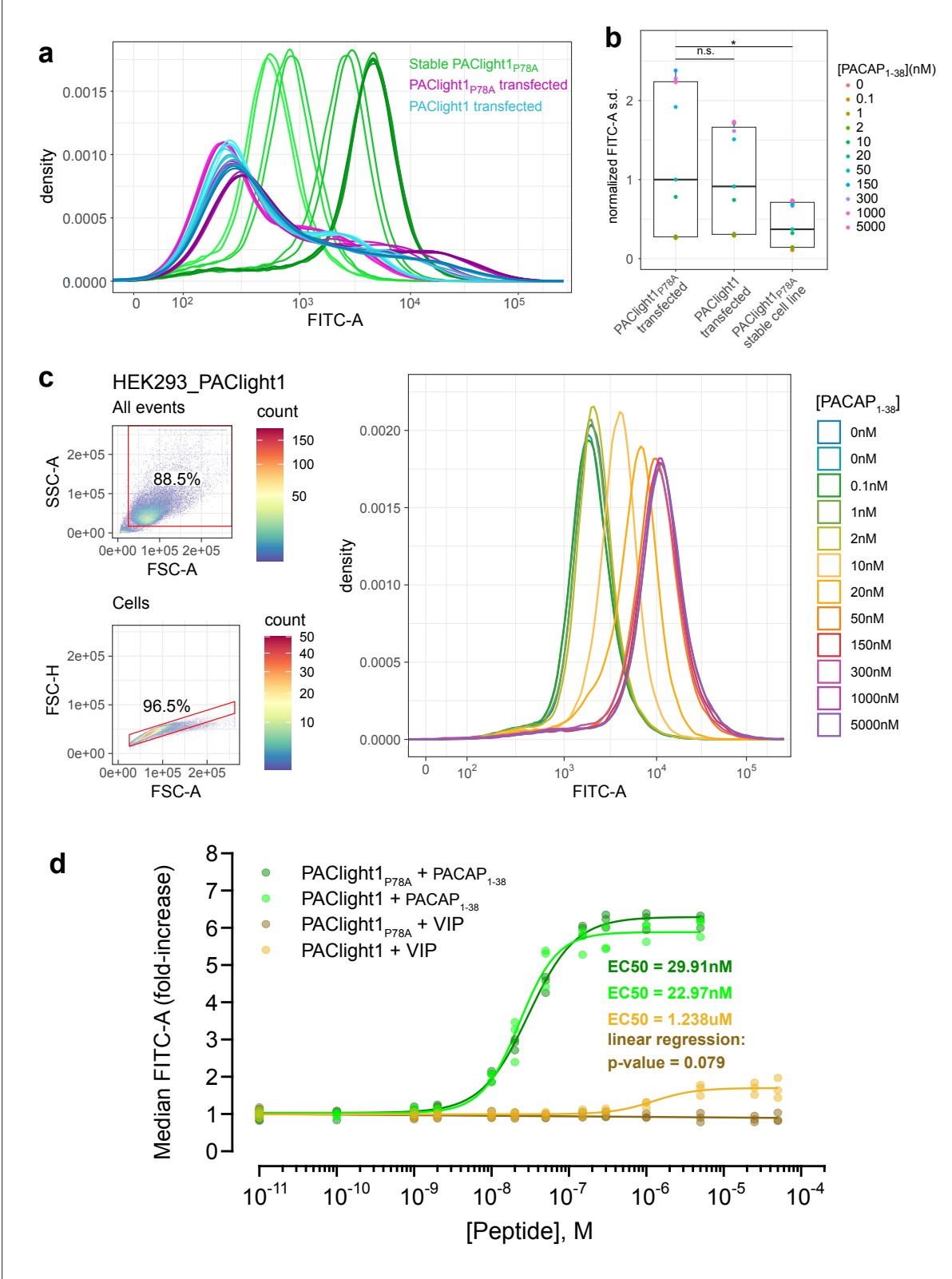

**Figure 3.** Comparison between transient and stable PAClight1$_{P78A}$ expression. (**a**) A direct flow cytometric comparison between transfected HEK293T cells and stable and inducible HEK293_PAClight1$_{P78A}$ cells highlights the improved homogeneity of the distribution of expression levels in the newly generated stable cell line. Note the increased number of cells on both the low and high extremes in the transfected populations over the stable cell line population. Data from a full PACAP$_{1-38}$ titration series for each condition are presented. The overall variability in expression levels in the PAClight1$_{P78A}$-

*Figure 3 continued on next page*

*Figure 3 continued*

and PAClight1-transfected condition is very similar. (**b**) The standard deviation of each density curve (normalized to the median standard deviation of the transfected PAClight1$_{P78A}$ condition) is plotted by condition and concentration. The median standard deviation of the stable HEK293_PAClight1$_{P78A}$ condition is reduced to only 37.4% of the transfected PAClight1$_{P78A}$ condition (Dunnett's test p = 0.0142). The median standard deviation varies little between transfected PAClight1$_{P78A}$ and PAClight1 cells (8.6% lower in PAClight1, Dunnett's test p = 0.5966). (**c**) A representative example of a PACAP$_{1-38}$ titration series on stable HEK293_PAClight1 cells. Top left: Gating strategy used to gate cells by the area of the side scatter (SSC-A) vs. the area of the forward scatter (FSC-A). Bottom left: Gating strategy used to gate singlets (within the cells gate) by the height of the forward scatter (FSC-H) vs. the FSC-A. Right: Density curves of the FITC-A channel obtained from the singlet gate. (**d**) Dose–response curves obtained from vasoactive intestinal peptide (VIP) and PACAP$_{1-38}$ titrations on stable HEK293_PAClight1$_{P78A}$ and stable HEK293_PAClight1 cells. PAClight1$_{P78A}$ and PAClight1 both have high affinities for PACAP$_{1-38}$ (PAClight1$_{P78A}$ EC50 = 29.91 nM, PAClight1 EC50 = 22.97 nM, *n* = 3). While PAClight1 still shows a response to higher concentrations of VIP (EC50 = 1.24 µM), PAClight1$_{P78A}$'s response to VIP is completely abolished up to concentrations of 50 µM VIP ($F_{(1, 33)}$ = 3.28, p = 0.079, adj. $R^2$ = 0.06). Data in a–c are derived from 100 K original events recorded for each concentration and construct. Data in d are derived from 100 K recorded events across *n* = 3 of each titration series. * indicates p < 0.05 for Dunnett's test. n.s. = not statistically significant.

The online version of this article includes the following source data for figure 3:

**Source data 1.** Source data for graphs and bar plots in *Figure 3*.

With the stable cells and their improved homogeneity of expression levels, we then determined the affinities of PAClight1$_{P78A}$ and PAClight1 toward PACAP$_{1-38}$ and VIP. For each peptide concentration of a titration series 100,000 events were acquired, gated for cell clusters and for singlets. The FITC-A median fluorescence intensity (MFI) of the events in the singlet gate were used for dose–response fitting. A representative PACAP$_{1-38}$ titration series replicate on stable HEK293_PAClight1 cells is shown in *Figure 3c*. The titration dataset reveals a slightly lower affinity of the PAClight1$_{P78A}$ sensor (EC50 = 29.91 nM) than the PAClight1 sensor (EC50 = 22.97 nM) toward PACAP$_{1-38}$ (*Figure 3d*). The affinity of the PAClight1 sensor (non-specific mutant) toward VIP (EC50 = 1.238 µM) is much lower than its affinity toward PACAP$_{1-38}$ (*Figure 3d*). This 100–1000× difference is well in accordance with previous reports on the differential affinities of VIP and PACAP$_{1-38}$ on the PAC1R (*Cauvin et al., 1990*; *Buscail et al., 1990*). We were not able to detect any response of PAClight1$_{P78A}$ to the addition of VIP up to a concentration of 50 µM (linear regression p = 0.079, *Figure 3d*), corroborating the specificity of the PAClight1$_{P78A}$ sensor. Taken together, these data show that the PAClight1$_{P78A}$ and PAClight1 sensors display very high affinity toward PACAP$_{1-38}$, with a slightly lower affinity of PAClight1 compared to PAClight1$_{P78A}$ as the result of the point mutation on the extracellular domain of PAClight1$_{P78A}$.

## Characterization of PAClight1$_{P78A}$ and PAClight1$_{P78A}$-ctrl in model organism systems

Since PAClight1$_{P78A}$ showed excellent expression and response properties in neuronal cell cultures, we next tested whether the sensor can be established as a tool to test ligand binding in intact neuronal circuits of mammalian model systems. Publicly available RNAseq databases (in situ hybridization atlas from the Allen Institute: https://alleninstitute.org/) and previous work (*Zhang et al., 2021*) show strong expression of PACAP receptor (Adcyap1r1) in the cerebral cortex and hippocampus of mice and humans, suggesting these structures as interesting candidates for drug targeting. We tested the expression, sensitivity, and specificity of PAClight1$_{P78A}$ in acute mouse brain slices that provide a physiological environment to examine the actions of pharmacological agents within specific brain areas (*Loryan et al., 2013*).

AAVs encoding the PAClight1$_{P78A}$ or PAClight1$_{P78A}$-ctrl sensor under control of a human synapsin-1 promoter were stereotactically injected into the neocortex and hippocampus of adult mice (*Figure 4a*). Four weeks later, virus expression in the injection site was validated by histology (*Figure 4b*). PAClight1$_{P78A}$ and PAClight1$_{P78A}$-ctrl were efficiently expressed in neuronal cell bodies, axons and dendrites (*Figure 4c*, *Figure 4—figure supplement 1*).

To investigate the sensor response dynamics and its sensitivity, acute brain slices expressing PAClight1$_{P78A}$ or PAClight1$_{P78A}$-ctrl were prepared and PACAP$_{1-38}$ was bath applied at concentrations ranging from 0 to 3000 nM. We observed a dose-dependent fluorescence increase to PACAP$_{1-38}$ using PAClight1$_{P78A}$, but not PAClight1$_{P78A}$-ctrl (*Figure 4d*). PAClight1$_{P78A}$ fluorescence increased by +6.8 and +17.5% $\Delta F/F_0$ when bath applying 300 and 3000 nM of PACAP$_{1-38}$, respectively (*Figure 4d*). Comparing PAClight1$_{P78A}$ fluorescence with PAClight1$_{P78A}$-ctrl fluorescence revealed a statistically significant difference when bath applying 3000 nM of PACAP$_{1-38}$ (*Figure 4d*).

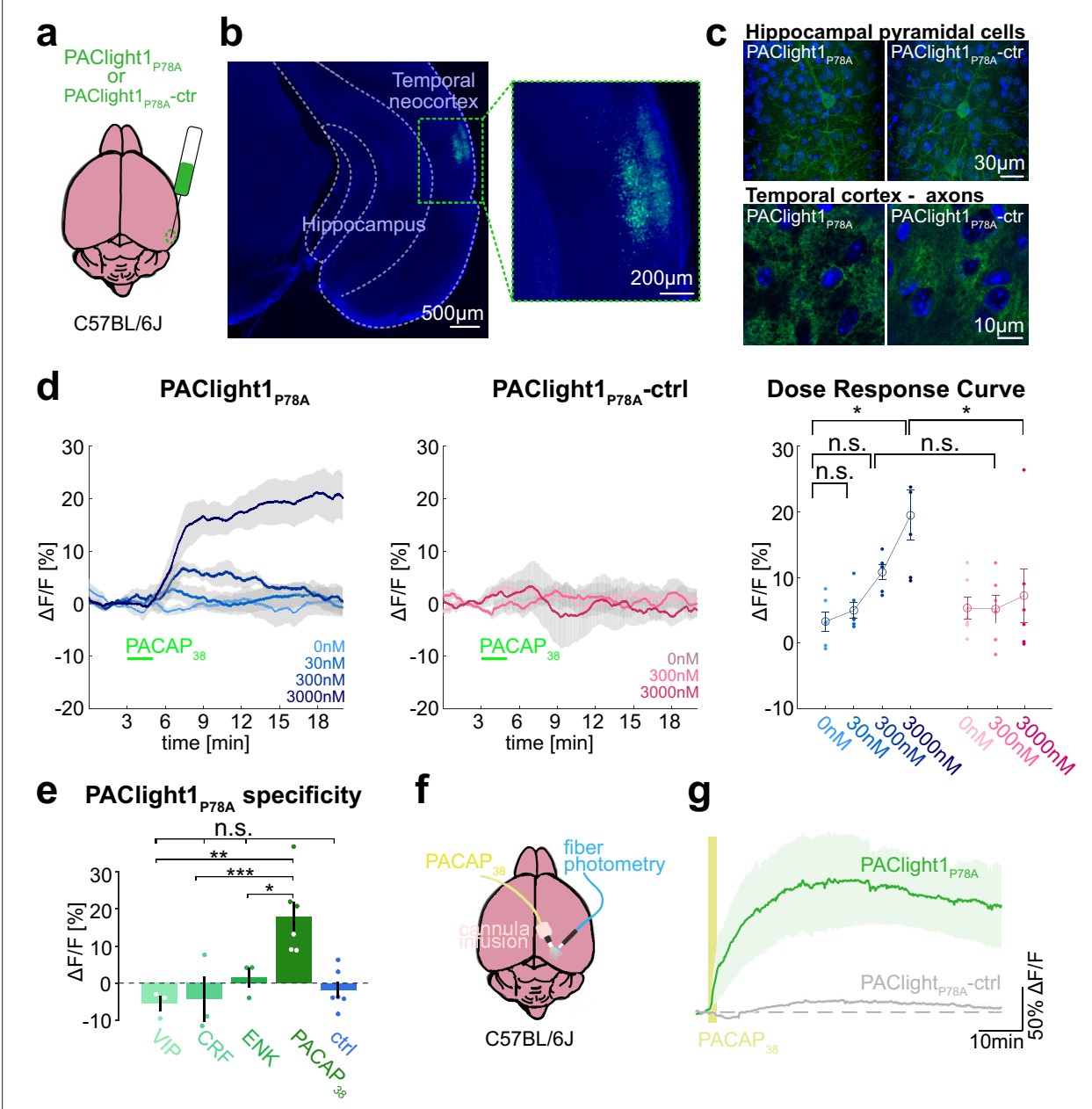

**Figure 4.** Ex vivo and in vivo sensor characterization in mice. (**a**) Adeno-associated viruses (AAVs) encoding the PAClight1$_{P78A}$ or PAClight1$_{P78A}$-ctrl sensor were injected into the temporal neocortex of adult mice. (**b**) Representative epifluorescent image of PAClight1$_{P78A}$ fluorescence after 4 weeks of expression time. (**c**) Maximum intensity projections of exemplary confocal images of PAClight1$_{P78A}$ and PAClight1$_{P78A}$-ctrl-expressing neurons and neuropil in hippocampus and cortex enhanced with GFP immunostaining and counterstained with DAPI (blue). (**d**) Acute mouse brain slices expressing PAClight1$_{P78A}$ (left) and PAClight1$_{P78A}$-ctrl (middle) were used to test the sensitivity of the sensor in the mammalian brain. PACAP$_{1-38}$ was bath applied for 2 min (green bar) at indicated concentrations. Data shown as mean ± standard error of the mean (SEM). $N$ = 6 slices per condition from ≥3 mice. Right: Dose–response curve for PAClight1$_{P78A}$ $\Delta F/F_0$ (blue shades) and PAClight1$_{P78A}$-ctrl $\Delta F/F_0$ (pink shades) in response to indicated concentrations of PACAP$_{1-38}$. Mann–Whitney $U$ tests with Bonferroni corrections revealed no statistically significant difference in mean PAClight1$_{P78A}$ peak responses to bath application of 30 nM (p = 2.909), and 300 nM PACAP$_{1-38}$ (p = 0.0519) compared to 0 nM PACAP$_{1-38}$. We found a significant difference between 0 and 3000 nM PACAP$_{1-38}$ (p = 0.013). Comparing mean peak $\Delta F/F_0$ PAClight1$_{P78A}$ responses with PAClight1$_{P78A}$-ctrl revealed a significant difference when bath applying 3000 nM (p = 0.041), but not for 300 nM (p = 0.084). (**e**) Quantification of mean PAClight1$_{P78A}$ peak responses to bath application of 3 μM vasoactive intestinal peptide (VIP), CRF, ENK, PACAP$_{1-38}$, and negative control (ctrl) in acute mouse brain slices. $N$ = 3–6 slices per condition from ≥3 mice. No statistically significant difference ($F_{(4,10)}$ = 5.15) was found between ctrl and VIP (p = 0.9705), ctrl and CRF (p = 0.9937) and ctrl and ENK (p = 0.9726). A statistically significant difference ($F_{(3,8)}$ = 9.19) was detected between PACAP$_{1-38}$ and VIP (p = 0.007), PACAP$_{1-38}$ and CRF (p = 0.001), and PACAP$_{1-38}$ and ENK (p = 0.045). Statistically significant differences are indicated with an asterisk, non-significant differences with n.s. (**f**) AAVs encoding

*Figure 4 continued on next page*

*Figure 4 continued*

the PAClight1$_{P78A}$ or PAClight1$_{P78A}$-ctrl sensor were injected into the neocortex of adult mice. Fiberoptic cannulae and acute microinfusion cannulae were implanted nearby. (**g**) PAClight1$_{P78A}$ and PAClight1$_{P78A}$-ctrl fluorescence changes upon microinfusion of 300 µM PACAP$_{1-38}$ (200 nl) recorded with fiber photometry in freely behaving mice. $N$ = 5 PAClight1$_{P78A}$ and 4 PAClight1$_{P78A}$-ctrl mice. Data shown as mean ± SEM.

The online version of this article includes the following source data and figure supplement(s) for figure 4:

**Source data 1.** Source data for graphs and bar plots in *Figure 4*.

**Figure supplement 1.** Validation of PAClight1$_{P78A}$ and PAClight1$_{P78A}$-ctrl sensors in mammalian brains.

To assess the specificity of PAClight1$_{P78A}$ to PACAP$_{1-38}$ in comparison to other neuropeptides in acute brain slices, we recorded its response to 3 µM VIP, corticotropin-releasing factor (CRF), and enkephalin (ENK). PAClight1$_{P78A}$ clearly increased its fluorescence in response to 3 µM PACAP$_{1-38}$ (+17.5% peak $\Delta F/F_0$), but not to VIP (−5.5% peak $\Delta F/F_0$), CRF (−4.3% peak $\Delta F/F_0$), or ENK (+1.4% peak $\Delta F/F_0$) (*Figure 4e*).

In conclusion, our data show that in acute mouse brain slices, PAClight1$_{P78A}$ detects concentrations of >300 nM of PACAP$_{1-38}$ in superfused bath application, while not reacting to other neuropeptides tested.

## In vivo PACAP detection in behaving mice

To test whether PAClight1$_{P78A}$ can be used to characterize ligand binding and diffusion in vivo in behaving mice, we implanted fiberoptic cannula into the neocortex of mice expressing PAClight1$_{P78A}$ or PAClight1$_{P78A}$-ctrl to image in vivo fluorescence dynamics while microinjecting PACAP$_{1-38}$ through nearby cannula (*Figure 4f*). Microinfusion of PACAP$_{1-38}$ (300 µM, 200 nl) led to peak fluorescence increases of 165.5 ± 58.0% $\Delta F/F_0$ in PAClight1$_{P78A}$-expressing mice. The fluorescence peaked at 28 min and dropped to 128.6 ± 45.7% $\Delta F/F_0$ 1 hr after, when injections were positioned in average 318 µm from the recording site. In contrast, PAClight1$_{P78A}$-ctrl-expressing mice showed a fluorescence increase of only 16.7 ± 2.6% $\Delta F/F_0$ (*Figure 4g*).

In conclusion, we show that PAClight1$_{P78A}$ can be a useful tool to detect PACAP$_{1-38}$ in vivo with a dynamic range that allows for the detection of drug injections even in the presence of intact endogenous PACAP systems in mice. Moreover, our data suggest that PACAP$_{1-38}$ diffuses efficiently across hundreds of µm, with slow extracellular degradation in neocortical brain areas of mice.

## Two-photon validation of the sensor in living zebrafish

Zebrafish larvae are an important animal model that has long been recognized for its utility and applicability to drug discovery (*MacRae and Peterson, 2015*; *Zon and Peterson, 2005*; *Sturtzel et al., 2023*). Moreover, PAC1 signaling has been associated with the adaptive stress response of zebrafish (*Biran et al., 2020*; *Amir-Zilberstein et al., 2012*). The high degree of amino acid sequence conservation of PACAP across the phylogenetic tree, as well as the in vitro response of PAClight1$_{P78A}$ to zebrafish PACAP that we observed, motivated us to also functionally validate our sensor for use in live zebrafish larvae. Based on a publicly available single-cell RNA sequencing (scRNAseq) dataset (*Farrell et al., 2018*), we identified the olfactory region of 4-day post-fertilization (dpf) old zebrafish larvae to express high levels of *Adcyap1b* in ~75% of cells composing the olfactory region. The presence of *Adcyap1b* expression in the olfactory region was further confirmed with data obtained by *Farnsworth et al., 2020*; *Figure 5a*. To induce expression of PAClight1$_{P78A}$ in the olfactory region, we used the gal4 driver line Tg(GnRH3:gal4ff), which can strongly drive expression (e.g., of GCaMP6s) in the olfactory bulbs (OBs; *Figure 5b*). In conjunction with this gal4 driver line, we used Tol2-mediated integration of a UAS-promoted PAClight1$_{P78A}$ construct. At 4 dpf, we immobilized the larvae with low-melting point agarose and performed two-photon volumetric imaging of the olfactory region for 5 consecutive 3D volumes as baseline. The immobilized zebrafish larvae were then placed onto a micromanipulator platform to inject 50 nl of a 1 mM PACAP$_{1-38}$ in saline solution or 50 nl of saline only (negative control) into the ventricular space (i.e., intracerebroventricular (ICV) injection). Subsequently, the larvae were re-imaged for 15 additional 3D volumes to record potential alterations in the pixel intensity values emitted by the PAClight1$_{P78A}$ sensor (*Figure 5c, c'*). We observed a strong increase in PAClight1$_{P78A}$ fluorescence already at ~120 s post-ICV injection with $\Delta F/F_0$ levels continuously rising until the end of the post-ICV injection recordings (*Figure 5d, d''*). Peak $\Delta F/F_0$ levels and the area under the curve

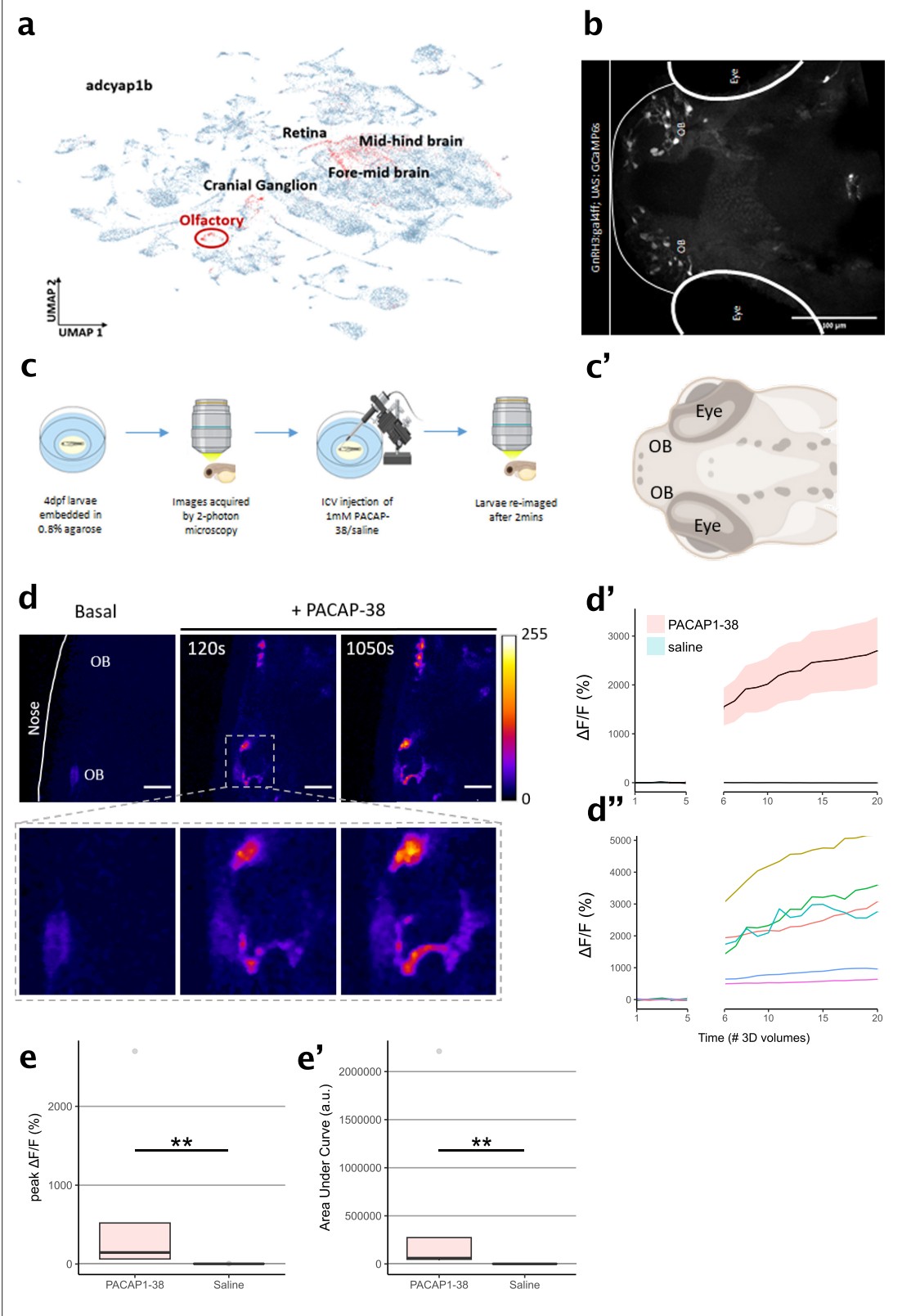

**Figure 5.** Characterization of PAClight1 in live zebrafish larvae. (**a**) Uniform manifold approximation and projection (UMAP) showing topological distribution of single-cell gene expression clusters with high adcyap1b (PACAP) expression of 0–4 dpf larvae. High adcyap1b expression is highlighted in the olfactory bulb. (**b**) Maximum intensity projection image of the 4 dpf *Tg(GnRH3:gal4ff; UAS:GCaMP6s)* larvae shows GnRH3-expressing cells in the olfactory bulb. (**c**) Schema representing the experimental design. 4 dpf larvae were immobilized and 3D volumetric images across time were

*Figure 5 continued on next page*

*Figure 5 continued*

obtained during the naïve state. 50 nl of 1 mM PACAP-38/saline was injected intracerebroventricularly and the same larvae was imaged again after 2 min. (**c'**) Schema of imaged region of interest including the olfactory bulb in the 4 dpf larvae. Illustrations in c and c' were created using Biorender. com. (**d**) (Top) Representative maximum intensity projection of two-photon volumetric images showing naïve sensor at baseline and its increase in fluorescence after PACAP-38 injection in the same larvae along time (scale bar = 20 μm). Color bar is representative of the $\Delta F/F_0$ in the images. (Bottom) Zoomed-in image of the left olfactory bulb showing increase in fluorescence after injection of PACAP$_{1-38}$. (**d'**) Representative quantification of the change in fluorescence with respect to basal fluorescence depicted as $\Delta F/F_0$. PACAP$_{1-38}$ injected larvae show an increase in activity of the fluorescent sensor as compared to their saline injected sibling controls. (**d''**) Individual cell traces show variability in sensor responses in different cells that may be a function of expression of the sensor on the cell surface. (**e, e'**) Peak $\Delta F/F_0$ and area under the curve (AUC) of PACAP$_{1-38}$ injected larvae is significantly higher than the saline injected controls ($n_{control}$ = 5, $n_{PACAP-38}$ = 5, Kolmogorov–Smirnov test p = 0.0079). ** indicates p < 0.01 for Kolmogorov–Smirnov test.

The online version of this article includes the following source data for figure 5:

**Source data 1.** Source data for graphs and bar plots in *Figure 5*.

---

significantly differed between PACAP$_{1-38}$/saline and saline-only injected animals ($n_{saline}$ = 5, $n_{PACAP1-38}$ = 5, Kolmogorov–Smirnov test p = 0.0079, *Figure 5e, e'*). The condition median for peak $\Delta F/F_0$ levels was 70-fold larger in PACAP$_{1-38}$/saline injected animals (145.76% $\Delta F/F_0$) than in saline-only injected animals (2.06% $\Delta F/F_0$) (*Figure 5e*). However, we noticed considerable variability in the extent of the dynamic range displayed by different regions of interest (ROIs) within the same and also between different zebrafishes (*Figure 5d'', e'*).

## Discussion

In this work, we engineered a new family of genetically encoded fluorescent sensors using the human PAC1R as a GPCR scaffold. Two of the sensors that we developed (PAClight1 and PAClight1$_{P78A}$) exhibit a very high dynamic range (above 1000% $\Delta F/F_0$), excellent expression at the cell surface, high basal brightness, and retain the pharmacological profile and ligand-binding profile of the parent receptor, with the exception of PAClight1$_{P78A}$ whose response to VIP is intentionally abolished via a single-point mutation. Given the very high sensitivity of these tools, future work could explore whether a grafting-based approach, similar to the one we recently described for class-A GPCR-based sensors (*Kagiampaki et al., 2023*), could lead to the direct generation of multiple class-B1 sensors based on the optimized fluorescent protein module from PAClight.

As part of this sensor family, we also introduced a control sensor harboring the two-point mutations R199A and Y161A, in which the response to PACAP$_{1-38}$ is abolished. Given the higher PACAP selectivity of PAClight1$_{P78A}$, this sensor variant is intended to be employed when there is an experimental need to ensure maximal selectivity of the response for the endogenous ligand PACAP$_{1-38}$ (e.g., if the sensor is to be used for attempting the detection of endogenous PACAP$_{1-38}$ release). For all other experimental scenarios, PAClight1 can be a better choice, as it retains the original wild-type sequence in its ECD and thus most closely represents the natural ligand-binding profile of the human PAC1 receptor.

Importantly, we demonstrate successful expression and functionality of PAClight1$_{P78A}$ and PAClight1$_{P78A}$-ctrl in the mouse brain by injecting AAVs encoding the sensor stereotactically. Additionally, our experiments in zebrafishes and mice revealed intriguing inter- and intraindividual differences in maximal sensor response to ICV and intracranial injections of the peptide ligand. These differences could potentially stem from variances in the expression density of PAClight1$_{P78A}$ within single cells. An intriguing alternative explanation, supported by the observation of continuously increasing $\Delta F/F_0$ levels (at least within the first 15 min), is that variations in diffusion rates and spatial diffusion patterns of PACAP$_{1-38}$ within the brain parenchyma and cortical tissue may contribute to these differences. Furthermore, the distance between PAClight1$_{P78A}$-expressing cells and the site of injection might influence the extent of fluorescent response. Under more controlled experimental conditions, PAClight1/ PAClight1$_{P78A}$-expressing zebrafishes and mice could therefore potentially be used to investigate and validate diffusion properties of PAC1R ligands in live organisms at a relatively high throughput or to

screen for novel small-molecule or peptide PAC1R agonists and antagonists with improved penetration properties across the ventricular wall.

Consistent with our findings in zebrafish, $PACAP_{1-38}$ microinjections into the mouse cortex revealed slow increases in $\Delta F/F_0$ levels even when $PACAP_{1-38}$ was injected only 100 s of μm away from the recording site, substantiating the hypothesis of slow diffusion of $PACAP_{1-38}$ through brain tissue. Furthermore, our results show a sustained activity with a slow decay of $PAClight_{P78A}$ fluorescence, especially at high concentrations. We hypothesize that this delayed decay may result from the slow clearance of peptides from extracellular space, putatively due to saturation of the enzymatic degradation system. An alternative explanation could be a long-lasting activation of PAC1 after binding of $PACAP_{1-38}$. This explanation is substantiated by slow decay in our slice experiments and by previous findings that $PACAP_{1-38}$ effects on downstream PKA signaling is longer lasting than effects of two other tested peptides (VIP and CRH) (*Hu et al., 2011*).

Notably, in this work, we did not demonstrate the use of any of our PAClight sensors for the detection of endogenous PACAP release in tissues or animal models, as our focus was deliberately set on the characterization and validation of the tools for applications to drug screening and/or development across species. Future work could focus on employing these tools to test whether or not they could be suitable for detecting endogenous PACAP dynamics with high spatiotemporal resolution in tissues or awake behaving animals. Based on our observations from mouse slice and in vivo recordings, bulk or epifluorescent imaging is unlikely to reveal endogenous release of PACAP, given that measured peptide concentrations typically occur in the femtomolar to nanomolar range (*Palkovits et al., 1995*). However, further optimization of the sensor to increase sensitivity and brightness coupled with the use of two-photon imaging that allows to focus on PAClight-expressing cell membranes nearby PACAP release sites, holds promise for future application of this sensor for in vivo detection of endogenously released PACAP. Considering the slow diffusion of $PACAP_{1-38}$ observed in our in vivo experiments, it will be quintessential to record PACAP release close to the main sites of action to capture physiologically relevant temporal dynamics. This could be achieved for example also by expressing the sensor exclusively in cells that express PACAP-sensitive receptors (PAC1, VPAC1, and VPAC2) and then trigger PACAP release via optogenetic stimulation or behavioral paradigms.

Our results show that the new tools introduced in this study can be valuable assets to investigate real-time dynamics of PAC1 receptor activation in response to the application of specific PAC1 receptor agents, both at a cellular level and in vivo in zebrafish and in the mammalian brain. The in vivo validation of our sensor demonstrates its ability to reveal diffusion dynamics of applied drugs and peptides within brain tissue. We therefore suggest that this tool can be employed in applications such as studying the diffusion of novel PAC1-targeting drugs and peptides, thereby offering mechanistic insights into the localization of drug actions in the brain.

# Methods

**Key resources table**

| Reagent type (species) or resource | Designation | Source or reference | Identifiers | Additional information |
|---|---|---|---|---|
| Strain, strain background (*Escherichia coli*) | NEB 10 Beta | New England Biolabs | C3019H | N/A |
| Cell line (*Homo sapiens*) | HEK293T | ATCC | CRL-3216 | N/A |
| Cell line (*Homo sapiens*) | Flp-In T-REx 293 cells | Thermo Fisher | R78007 | N/A |
| Commercial assay or kit | Nano-Glo Live Cell Reagent | Promega | N2011 | N/A |
| Commercial assay or kit | Lipofectamine 2000 | Thermo Fisher | 11668019 | N/A |
| Commercial assay or kit | Effectene | QIAGEN | 301425 | N/A |
| Chemical compound, drug | $PACAP_{1-38}$ | Sigma-Aldrich | A1439 | N/A |
| Software, algorithm | Fiji | ImageJ | 2.15.1 | https://imagej.net/software/fiji/ |

*Continued on next page*

| Reagent type (species) or resource | Designation | Source or reference | Identifiers | Additional information |
|---|---|---|---|---|
| | *Continued* | | | |
| Software, algorithm | R studio | Posit | 2023.09.1 | https://posit.co/downloads/ |

## Molecular cloning

A synthetic DNA geneblock for the human PAC1R-null receptor sequence (PAC1R) was designed and ordered (Life Technologies) based on the NCBI protein data bank entry 'NP_001109.2'. The protein-coding sequence was codon optimized and flanked by HindIII (5′) and NotI (3′) restriction sites for cloning into a pCMV plasmid (RRID:Addgene _60360). A hemagglutinin signal peptide (MKTIIALSYIF-CLVFA) was introduced 5′ to the PAC1R-coding sequence. Circular Polymerase Extension Cloning was used to replace the intracellular loop 3 (Q336–G342) of PAC1R with a cpGFP module from dLight1 (*Patriarchi et al., 2018*). For sensor optimization, libraries of sensor variants were created through site-directed mutagenesis. For signaling assays, a linker + SmBit sequence (*Laschet et al., 2019*) (GNSGSSGGGGSGGGGSSGG + VTGWRLCERILA) was cloned at the 3′end of both the PAClight1$_{P78A}$ and the PAC1R sequence. The XhoI cleavage site from within the SmBit linker sequence and the first four residues of the SmBit linker (GNSG) were first cloned into the 3′end of the PAClight1$_{P78A}$ and PAC1R sequences using PCR. Subsequently, a restriction digest using XhoI and XbaI restriction enzymes was performed on the SmBit containing plasmid (B2AR-SmBit) as well as on the pCMV_PAClight1$_{P78A}$ and pCMV_PAC1R plasmids. Ligation of the linker + SmBit insert into the linearized backbones was performed after gel extraction and purification of the insert. For generation of stably expressing and inducible Flp-In T-REx 293 cells, the PAClight1$_{P78A}$ and PAClight1 sequences were cloned into the pcDNA5/FRT/TO backbone, respectively, using restriction digest with BamHI and NotI restriction enzymes. PCR reactions were performed using a Pfu-Ultra II Fusion High Fidelity DNA Polymerase (Agilent), whereas Gibson Assembly was performed using NEBuilder HiFi DNA Assembly Master Mix (New England Biolabs). All sequences were verified using Sanger sequencing (Microsynth).

## Structural modeling, protein sequence alignment, and peptide synthesis

Modeling of protein structure for the PAClight1$_{P78A}$ sensor construct was performed using AlphaFold2 (*Mirdita et al., 2022*), using pdb70 as a template mode. The best-scoring prediction was then manually edited using UCSF Chimera (version 1.13.1). Multiple sequence alignment was performed using ClustalOmega (*Sievers and Higgins, 2014*) and visualized with the Jalview (*Waterhouse et al., 2009*) software (version 2). Zebrafish PACAP2$_{1-27}$(zfPACAP2), chicken PACAP$_{1-38}$ (chPACAP38), amnesiac, maxadilan (Maxadilan$_{1-61}$), and Max.d.4 peptides were synthesized on an automated fast-flow peptide synthesizer using a previously described protocol (*Hartrampf et al., 2020*).

## Cell culture, imaging, and quantification

HEK293T cells (ATCC #CRL-3216) were cultured in DMEM medium (Thermo Fisher) supplemented with 10% fetal bovine serum (FBS; Thermo Fisher) and 1× Antibiotic–Antimycotic (100 units/ml of penicillin, 100 µg/ml of streptomycin, and 0.25 µg/ml of Amphotericin B, Thermo Fisher) and incubated at 37°C with 5% $CO_2$. Cells were transfected at 50–60% confluency in glass-bottomed dishes using the Effectene transfection kit (QIAGEN) according to the manufacturer's instructions, and imaged 24–48 hr after transfection, as previously described (*Duffet et al., 2022*). Primary cultured hippocampal neurons were prepared and transduced as previously described (*Kagiampaki et al., 2023*). Before imaging, all cells were rinsed with 1 ml of Hank's Balanced Salt Solution (Life Technologies) supplemented with $Ca^{2+}$ (2 mM) and $Mg^{2+}$ (1 mM). Time-lapse imaging was performed at room temperature (22°C) on an inverted Zeiss LSM 800 confocal microscope using either a ×40 oil-based or a ×20 air objective. Longer-term imaging for determination of internalization/stability as well as reversibility was performed using a ×10 air objective, with 1 frame (1024 × 1024 pixels) acquired every minute (2× pixel averaging). Imaging was performed using a 488-nm laser as excitation light source for PAClight1 sensors. During imaging, ligands were added in bolus on the cells using a micropipette to reach the final specified concentrations of ligands on the cells. For quantification of recordings in single dishes, an average intensity projection across the whole temporal stack was first generated. Pixel-intensity-based thresholding in Fiji was then performed manually on the average intensity

projection to segment the plasma membrane as accurately as possible. Using the magic wand tool from Fiji, connected patches of segmented plasma membranes (sampled from multiple cells) were selected as ROIs. The ROIs were then projected onto the temporal stack to make sure the plasma membrane did not drift out of the ROI during the time lapse recording. Sensor response ($\Delta F/F_0$) was calculated as the following: $(F(t) − F\_base)/F\_base$ with $F(t)$ being the ROI fluorescence value at each time point ($t$), and $F\_base$ being the mean fluorescence of the 10 time points prior to ligand addition.

## Stable cell line generation and maintenance

The stable cell line for tetracycline-inducible expression of PAClight1$_{P78A}$ was generated following previously described procedures (*Klein Herenbrink et al., 2022*). Flp-In T-REx 293 cells were grown in DMEM supplemented with 10% (vol/vol) FBS (Invitrogen), 100 µg/ml zeocin (Thermo Fisher Scientific), and 15 µg/ml blasticidin (Thermo Fisher Scientific). To obtain a cell line with stably integrated PAClight1$_{P78A}$ or PAClight1 expression cassettes, cells were grown in T150 flasks (Corning) until 70% confluency, were then transfected with 0.6 µg of pcDNA5/FRT/TO-PAClight1$_{P78A}$ DNA vector and 5.4 µg pOG44 vector using Effectene transfection kit (QIAGEN). Two days after transfection, cells were split and the medium was changed to DMEM supplemented with 10% (vol/vol) FBS, 200 µg/ml Hygromycin B (Sigma), and 15 µg/ml blasticidin. The medium was then replaced twice a week until individual colonies were visible. An individual colony was manually selected and expanded for subsequent experiments. Induction of PAClight1$_{P78A}$ or PAClight1 expression was obtained by adding 1 µg/ml doxycycline (Sigma) to the cell medium 1–2 days prior to experimentation.

## Spectral characterization of the sensors

One-photon spectral characterization of the PAClight1$_{P78A}$ sensor was performed using PAClight1$_{P78A}$-transfected HEK293T cells before and after addition of PACAP$_{1-38}$ (10 µM). One-photon fluorescence excitation ($\lambda_{em}$ = 560 nm) and emission ($\lambda_{exc}$ = 470 nm) spectra were determined on a Tecan M200 Pro plate reader at 37°C. 24 hr after cell transfection in 6-well format (linear PEI), ~1 million cells were dissociated with addition of TrypLE Express (Thermo Fisher) and thoroughly washed with phosphate-buffered saline (PBS). Next, cells were resuspended in 300 µl of PBS and aliquoted into two individual wells of a 96-well microplate with or without PACAP$_{1-38}$ (10 µM), together with two wells containing the same amount of non-transfected cells to account for autofluorescence and a single well containing PBS for subtraction of the Raman bands of the solvent.

## Flow cytometry

HEK293T cells and stable HEK293_PAClight1$_{P78A}$/PAClight1 cells were seeded into T175 flasks and grown to 50–60% confluency under culture conditions described above. HEK293T cells were then transfected with 20 µg of pCMV_PAClight1$_{P78A}$ or pCMV_PAClight1 using linear PEI (Sigma-Aldrich; #764965) and a PEI-to-plasmid ratio of 3:1. Stable HEK293_PAClight1$_{P78A}$/PAClight1 cells were induced with 1 µg/ml doxycycline (Sigma) at 50–60% confluency. Two days after transfection or induction, cells were washed in 1× PBS before detachment with TrypLE Express (Thermo Fisher) and subsequent centrifugation at 150 × *g* for 3 min. Palleted cells were then resuspended in ice-cold FACS buffer containing 1× PBS, 1 mM EDTA, 25 mM HEPES (pH 7.0), 1% FBS and diluted to 4–4.8 × $10^6$ cells/ml. A dilution series of PACAP$_{1-8}$ and VIP was prepared in FACS buffer. 125 µl of peptide solution was pipetted into a 96-well format and another 125 µl of cell suspension was added to each well. The 96-well U-bottom plate was then loaded into a BD FACS Canto II cytometer equipped with a high-throughput sampler for 96-well format sampling. Voltage for the photomultiplier tubes was set to 200 V for the forward scatter detector, to 400 V for the side scatter detector, and to 350 V for the FITC channel fluorescence detector. Excitation was performed at 488 nm, while emission was directed through a 502LP mirror and a 530/30 band pass filter. Sampling from 96-well format wells was performed with an initial mixing step (3 × 100 µl mixes at 150 µl/s) followed by sample acquisition of 200 µl at a flow rate of 1–3 µl/s. Sampling was performed until 100 K events were recorded per well. After sample acquisition of each well, a 800-µl wash step of the sampling line was performed. Sample acquisition within a dilution series replicate was performed from low-to-high concentrations to minimize potential peptide carry-over into the neighboring wells. Furthermore, a minimum of six wells without PACAP or VIP present were sampled between each replicate of the same construct. Raw data were exported in FCS3.1 format and further processed for analysis within R. The following

R packages were used for the analysis and visualization of the flow cytometry experiments: flowCore, flowAI, ggcyto, dplyr, tidyr, forcats, purrr, stringr, openCyto, openxlsx, svglite, ggprism, ggnewscale, and glue. After import of the raw datasets, a rectangular gate (Cells) was defined on a FSC-A vs. SSC-A scatter plot with the limits being set to 25K- and 15K-infinity, respectively. Next, a 'Singlet' gate within the parent gate 'Cells' was created automatically using the openCyto::singletGate function based on the FSC-A vs. FSC-H scatter plot. MFIs of the FITC-A channel for all recorded events within the 'Singlet' gate were calculated for each well. The MFI values were then normalized (NormFITC_A) within each titration replicate to the average of the 0 nM peptide condition. Dose–response curves and $EC_{50}$ values were obtained by first grouping datasets by construct and peptide conditions. For each group, a non-linear least squares model was fit using the following formula:

$$NormFITC\_A \sim min + ((max - min)/(1 + exp(hill\_coefficient * (EC50 - Conc\_M))))$$

For the VIP titration on the non-responsive sensor variant, a linear regression model was fit using the following formula:

$$NormFITC\_A \sim Conc\_M$$

## NanoLuciferase complementation assays

HEK293 cells were seeded in 6-well plates and transfected with 0.25µg LgBiT-miniG (miniGs or miniGsq; *Wan et al., 2018*) or LgBiT-β-arrestin2 (*Laschet et al., 2019*) and 0.25 µg SmBiT-tagged receptor plasmids using 3 µl Lipofectamine 2000 (Thermo Fisher). 24 hr after transfection, cells were plated onto black clear-bottomed 96-well plates at 50,000–100,000 cells/well or 384-well plates at 20,000–30,000 cells/well in DMEM without phenol red, supplemented with 30 mM HEPES (pH 7.4) and Nano-Glo Live Cell Reagent (Promega), and incubated for 45 min at 37°C. Luminescence signal was measured simultaneously across the plate using the FDSS/µCELL plate reader (Hamamatsu). Baseline luminescence (before agonist addition) was acquired for 3 min. Vehicle (buffer) or agonist (PACAP$_{1-38}$, 1 µM) were added simultaneously into wells by an integrated dispensing unit. Luminescence was recorded every 1–2 s for 8 min post-agonist addition. Agonist-treated wells were initially normalized to vehicle wells, luminescence intensity was then normalized to the baseline prior to ligand addition.

## Generation and imaging of transgenic fishes

The PAClight1$_{P78A}$ sequence was cloned into a Tol2 transposase-based plasmid (*Kawakami, 2007*) with a UAS promoter. The plasmid (75 ng/µl) was injected into *Tg(GnRH3:gal4ff)* (*Golan et al., 2021*) embryos at the single-cell stage along with transposase mRNA (25 ng/µl). Embryos were sorted at 3 days post-fertilization using the red heart marker (cmlc:mCherry) in the *UAS:PAClight1$_{P78A}$* construct. Positive zebrafish larvae were immobilized using alpha-bungarotoxin (1 mg/ml for 30 s) and mounted ventrally (ventral side down) on to a custom molded 3 cm plate using 0.8% low-melt agarose (Sigma-Aldrich). Plates were perfused with E3 media (5 mM NaCl, 0.17 mM KCl, 0.33 mM CaCl$_2$, 0.33 mM MgSO$_4$ in H$_2$O) prior to imaging. Larvae were imaged under the Leica TCS SP8 Multi-Photon Microscope with a ×25/0.95 NA water immersion objective. Volumetric images across time were obtained after irradiation at 890 nm at 400 Hz/frame with 2× averaging. Larvae were imaged for five volumes in the naive state followed by ICV injections of 1 mM PACAP-38 (Sigma-Aldrich) in saline solution (0.6% NaCl, 0.02% Na$_2$CO$_3$) and only saline for controls. Larvae were then re-imaged approximately 120 s after ICV injection using the same imaging parameters for 15 volumes. Basal and post-injection images were concatenated and registered using descriptor-based series registration on ImageJ/Fiji (https://github.com/fiji/Descriptor_based_registration; *Preibisch et al., 2022*; *Schindelin et al., 2012*). Raw fluorescence traces were extracted from these images from manually drawn ROIs around cells. Statistical analysis and plotting of graphs were performed in MATLAB and R. Baseline fluorescence, $F_0$, was computed using basal fluorescence intensities, which were then used to calculate $\Delta F/F_0$. The Kolmogorov–Smirnov *t* test was performed where applicable. Single-cell transcriptomic data (*Farnsworth et al., 2020*) from 4 dpf zebrafish larvae was plotted as a function of uniform manifold approximation and projections in the UCSC cell browser. High expression of adcyap1b (PACAP) was visualized in Cluster 94 (here). Tg(GnRH3:gal4ff) transgenic line was used to express the PAClight1$_{P78A}$ in the OB.

To confirm that GnRH3 cells were indeed present in the OB at 4 dpf, a UAS:GCaMP6s construct was expressed in the Tg(GnRH3:gal4ff) line.

## Virus production

The biosensor AAV constructs were cloned in the Patriarchi laboratory, while the opsin AAV construct was constructed by the Viral Vector Facility of the University of Zürich (VVF). All viral vectors were produced by the VVF. The viral titers of the viruses used in this study were: AAV9.hSyn.PAClight1$_{P78A}$, 0.75–1.5 × 10$^{13}$ GC/ml, AAV9.hSyn.PAClight1$_{P78A}$-ctrl, 1.6 × 10$^{13}$ GC/ml.

## Animals

Rat embryos (E17) obtained from timed-pregnant Wistar rats (Envigo) were used for preparing primary hippocampal neuronal cultures. Wild-type C57BL/6JRj mice (Janvier, 6–10 weeks old) of both sexes were used in this study. Mice were kept in standard enriched cages with ad libitum access to chow and water on either normal or reversed 12/12 hr light/dark cycle. Mice were housed in cages of two to five animals.

All procedures in mice were performed in accordance with the guidelines of Medical University of Vienna and under approved licenses by the Austrian Ministry of Science.

Zebrafish were maintained and bred by standard protocols and according to FELASA guidelines. All experiments using zebrafish were approved by the Weizmann Institute's Institutional Animal Care and Use Committee (Application number 06340722-1).

## Stereotactic surgeries

For ex vivo slice imaging and validation of mouse brain expression, 6- to 10-week-old male and female C57BL/6JRj mice (Janvier) were anesthetized with 5% isoflurane and maintained under stable anesthesia during the surgery (1.3–2% isoflurane). Lidocaine (3.5 mg/kg; Gebro Pharma, #100562 1404) and Carprofen (4 mg/kg; Zoetis, #256684) were administered subcutaneously for local anesthesia and general analgesia. A small craniotomy was made bilaterally, targeting the temporal neocortex (Injection site at −4.1 mm posterior to bregma, as lateral as possible and 1.2 mm ventral to the pial surface). A glass micropipette was used for virus delivery and inserted slowly into the brain. Before and after virus delivery, the micropipette was kept in place for a waiting time of 8 min. 100–200 nl of AAV9.hSyn.PAClight1$_{P78A}$ or AAV9.hSyn.PAClight1$_{P78A}$-ctrl were injected, with an injection rate of 40 nl/min using a microsyringe pump (KD Scientific; #788110). The pipette was withdrawn slowly and the skin of the skull was sutured. The mice were allowed to recover for 4 weeks before being used for the experiments, with Carprofen (4 mg/kg) being administered on the 2 consecutive days after the surgery.

## Slice imaging

The mice were deeply anesthetized with isoflurane before being sacrificed by a transcardial perfusion with 30 ml cooled sucrose solution (212 mM sucrose, 3 mM KCl, 1.25 mM Na$_2$H$_2$PO$_4$, 26 mM NaHCO$_3$, 1 mM MgCl$_2$, 0.2 mM CaCl$_2$, 10 mM glucose) oxygenated with carbogen gas. The mice were decapitated and the brain was dissected out. 300 μM coronal slices were prepared using a Leica VT1200 vibratome in cold oxygenated sucrose solution (0.12 mm/s, 0.8 mm amplitude). The slices were incubated in oxygenated Ringer's solution (125 mM NaCl, 25 mM NaHCO$_3$, 1.25 mM NaH$_2$PO$_4$, 2.5 mM KCl, 2 mM CaCl$_2$, 1 mM MgCl$_2$, 25 mM glucose) at 37°C for 10 min and afterwards kept at room temperature. The slices were imaged using an Olympus BX51WI microscope with an ORCA-Fusion Digital CMOS camera (Hamamatsu, #C14440-20UP) and a ×10 objective (UMPLFLN10XW objective, Olympus). An EGFP(green fluorescent protein) filterset was used to control emission and excitation spectra (470/40, 525/50; AFH #F46-002). Micromanager software (*Edelstein et al., 2014*) was used to synchronize the excitation LED with the camera shutter. Videos were taken at 4 Hz framerate using 100ms exposure time per frame. Cortical areas with clearly visible baseline fluorescence were selected for imaging. Baseline recordings of 10–20 min were performed prior to data acquisition to allow the slices to accustom to the conditions. Bath temperature was set to ~31 °C. The experimental recording was 20 min long, with the respective peptide being infused from 3 to 5 min. The following peptides were used: PACAP$_{1-38}$ (30–3000 nM; Phoenix Pharmaceuticals; #052-05), VIP (3000 nM; Phoenix Pharmaceuticals; #064-16), CRF (3000 nM; Phoenix Pharmaceuticals; #019-06), and Met-Enkephalin (3000 nM; Phoenix Pharmaceuticals; #024-35). Of note, since our preliminary data

suggested a decrease in PAClight1$_{P78A}$ fluorescence in response to 3 µM CRF, we enriched the Ringer's solution with synaptic transmission and neuronal activity blockers (NBQX, CPP, CGP 55845, Gabazine, and TTX) to prevent recruitment of neuronal circuits that could lead to endogenous PACAP$_{1-38}$ release (Tocris, biotechne; #Tocris 1262/50, #Tocris 0373/50, #Tocris0247/50, #Tocris 1248/50, #Tocris 1069/1).

Data analysis was performed using a custom-written Matlab script. Movement correction as well as bleaching correction was performed. The average fluorescence across the whole imaging window was normalized to the baseline fluorescence measured during the first 3 min of the recording prior to peptide infusion to calculate $\Delta F/F_0$: $(F(t) - F\_base)/F\_base$ with $F(t)$ being the fluorescence value at each time point ($t$), and $F_0\_base$ being the mean fluorescence of the first 3 min. Six slices from at least three different mice were averaged for calculating the dose–response curve. For PAClight1$_{P78A}$ responses to VIP, CRF, and ENK, three slices from three different mice were averaged.

## In vivo photometric imaging and microinfusion

Craniotomies were made as described for stereotactic injections. Tapered fiberoptic cannula implants (MFC_200/230–0.37_2mm_MF1.25_A45) with low autofluorescence epoxy were implanted into the right neocortex at 0.9 mm depth with the angled (uncoated) side of the fiber tip facing toward the left. The implant was stabilized with cyanoacrylate glue.

Cannula were surgically implanted to locally infuse PACAP$_{1-38}$ into the cortex. Stainless steel guide cannulae (26 gauge; C315GA/SPC, InVivo One, Roanoke, VA, USA) were positioned in cortical L1, such that the tip of the internal cannula terminated 400 µm medial to an the fiberoptic cannula. The implant was secured with cyanoacrylate glue. Dummy cannulae that did not extend beyond the guide cannulae (C315DC/SPC, InVivo One) were inserted to prevent clogging.

## Mice were handled for 7 days prior to drug infusions

For photometric recordings, a 200-µm diameter and 0.37 NA patchcord (MFP_200/220/900–0.37_5m_FCM-MF1.25, low autofluorescence epoxy, Dorics) were used to connect the implanted fiberoptic cannulae to a Dorics filter cube for blue (465–480 nm) excitation light. Emission light was detected by built-in photodetectors (500–540 nm). Signals from the photodetectors were amplified with Dorics built-in amplifiers and acquired using a Labjack (T7). LJM Library (2018 release) was used to allow communication between MATLAB and Labjack. The voltage output from the LED drivers was amplitude modulated at 171 Hz (sine wave) as described previously (*Melzer et al., 2020*). Amplitude modulation was programmed in MATLAB. 470 nm LEDs (M470F3, Thorlabs; LED driver LEDD1B, Thorlabs) were used. Light power at the patchcord tip was set to an average of 45 µW. Fluorescence was calculated with custom-written MATLAB scripts based on a previous publication (*Owen and Kreitzer, 2019*). Photometry data were sampled at 2052 Hz.

For drug infusions, dummy cannulae were replaced by internal cannulae (33 gauge; C315LI/SPC, InVivo One) that extended 1 mm beyond the guide cannulae and were connected to a microinfusion pump. After a recovery time of 5–10 min, PACAP$_{1-38}$ (200 nl; 300 µM diluted in normal Ringer solution, NRR) was infused at 100 nl/min (NRR in mM: 135 NaCl, 5.4 KCl, 5 HEPES, 1.8 CaCl$_2$, pH 7.2 adjusted with KOH, sterile filtered with 0.2 µm pore size).

## Histology

To verify expression of the sensor in the mouse cortex and hippocampus, virus injections were performed with stereotactic surgeries as described above. The virus was allowed to express for 4 weeks, before the mice were transcardially perfused with PBS and 4% paraformaldehyde, before the brain was dissected and processed into 50 µm coronal slices using a Leica VT1000S vibratome. Slices were immunostained with chicken anti-GFP antibodies (Abcam Cat# ab13970, RRID:AB_300798; 1:500 diluted). In brief, slices were permeabilized and blocked for 1 hr with PBS containing 5% NGS(next-generation sequencing) and 0.2% Triton X-100, followed by a 24-hr antibody incubation at 4°C in PBS containing 5% NGS and 0.2% Triton X-100. Sections were washed in PBS and then incubated for 1 hr in 1:500 diluted goat anti-chicken IgG (H+L) cross-adsorbed secondary antibody (Alexa fluor 488; Fisher Scientific A11039). The mounted slices were imaged using a Leica SP8 X confocal microscope equipped with a ×63 1.3 NA glycerol immersion objective.

## Statistical analysis

For pairwise analysis of sensor variants, the statistical significance of their responses was determined on a case-by-case basis using a two-tailed unpaired Student's $t$ test with Welch's correction. ANOVA testing was followed by pairwise comparison with correction for multiple comparison with either Dunnett's correction or Hochberg correction. For data not meeting assumptions of normality or homoscedasticity, Mann–Whitney $U$ tests followed by Bonferroni corrections were applied. All p values are indicated either in the results section or in the figure legends. Data of sensor screening and validation experiments are displayed as mean ± 1 standard deviation. No statistical methods were used to predetermine sample size in cultured cells. Power calculations were performed to determine sample size for experiments using mice.

## Acknowledgements

The results are part of a project that has received funding from the European Research Council (ERC) under the European Union's Horizon 2020 research and innovation program (Grant agreement No. 891959) (TP). We also acknowledge funding from the Swiss National Science Foundation (Grant No. 310030_196455 and 310030L_212508), Olga Mayenfisch Foundation, and Hartmann Müller Foundation for Medical Research (TP). PR is supported by a research grant for student's fellowship from the Benoziyo Endowment Fund for the Advancement of Science and by the Weizmann–CNRS Collaboration Program. GL lab is supported by the Israel Science Foundation (#349/21); Israel Ministry of Science and Technology (#3-16548) and Hedda, Alberto, and David Milman Baron Center for Research on the Development of Neural Networks. MS and KA are supported by a Swiss National Science Foundation Eccellenza Professorial Fellowship to M.S. (PCEFP3_181282). SM and SN have been funded by the Vienna Science and Technology Fund (WWTF) and the City of Vienna through project VRG21-015. We would like to thank Prof. Marco Celio for generously contributing financial support during the project, Ulrik Gether (University of Copenhagen) for kindly providing us the Flp-In T-REx inducible cell complete system, as well as J-C Paterna and the Viral Vector Facility of the Neuroscience Center Zurich (ZNZ) for help with virus production.

## Additional information

### Competing interests

Tommaso Patriarchi: TP is a co-inventor on a patent application related to the technology described in this article. The other authors declare that no competing interests exist.

### Funding

| Funder | Grant reference number | Author |
|---|---|---|
| Olga Mayenfisch Stiftung | | Tommaso Patriarchi |
| European Research Council | 891959 | Tommaso Patriarchi |
| Schweizerischer Nationalfonds zur Förderung der Wissenschaftlichen Forschung | 310030_196455 | Tommaso Patriarchi |
| Schweizerischer Nationalfonds zur Förderung der Wissenschaftlichen Forschung | 310030L_212508 | Tommaso Patriarchi |
| Israel Science Foundation | 349/21 | Gil Levkowitz |
| Israel Ministry of Science and Technology | 3-16548 | Gil Levkowitz |

| Funder | Grant reference number | Author |
|---|---|---|
| Schweizerischer Nationalfonds zur Förderung der Wissenschaftlichen Forschung | PCEFP3_181282 | Miriam Stoeber |
| Vienna Science and Technology Fund (WWTF) | 10.47379/VRG21015 | Sarah Melzer |
| City of Vienna | VRG21-015 | Sarah Melzer |
| Benoziyo Endowment Fund for the Advancement of Science | | Preethi Rajamannar |

The funders had no role in study design, data collection, and interpretation, or the decision to submit the work for publication.

## Author contributions

Reto B Cola, Conceptualization, Resources, Data curation, Software, Formal analysis, Funding acquisition, Validation, Investigation, Visualization, Methodology, Writing – original draft, Writing – review and editing; Salome N Niethammer, Data curation, Formal analysis, Investigation, Visualization, Methodology, Writing – review and editing; Preethi Rajamannar, Data curation, Formal analysis, Investigation, Visualization, Methodology; Andrea Gresch, Musadiq A Bhat, Elyse T Williams, Patrick Hauck, Nina Hartrampf, Dietmar Benke, Investigation, Methodology; Kevin Assoumou, Formal analysis, Investigation, Visualization, Methodology; Miriam Stoeber, Investigation, Visualization, Methodology; Gil Levkowitz, Resources, Software, Formal analysis, Supervision, Investigation, Visualization, Methodology, Project administration, Writing – review and editing; Sarah Melzer, Resources, Data curation, Software, Formal analysis, Supervision, Validation, Investigation, Visualization, Methodology, Writing – original draft, Writing – review and editing; Tommaso Patriarchi, Conceptualization, Resources, Data curation, Software, Formal analysis, Supervision, Funding acquisition, Validation, Investigation, Visualization, Methodology, Writing – original draft, Project administration, Writing – review and editing

## Author ORCIDs

Reto B Cola ⬤ https://orcid.org/0000-0003-1419-2480
Musadiq A Bhat ⬤ https://orcid.org/0000-0002-0894-9996
Nina Hartrampf ⬤ https://orcid.org/0000-0003-0875-6390
Miriam Stoeber ⬤ https://orcid.org/0000-0002-5210-2864
Gil Levkowitz ⬤ https://orcid.org/0000-0002-3896-1881
Sarah Melzer ⬤ https://orcid.org/0000-0001-6028-9764
Tommaso Patriarchi ⬤ https://orcid.org/0000-0001-9351-3734

## Ethics

All procedures in mice were performed in accordance with the guidelines of Medical University of Vienna and under approved licenses by the Austrian Ministry of Science. Zebrafish were maintained and bred by standard protocols and according to FELASA guidelines. All experiments using zebrafish were approved by the Weizmann Institute's Institutional Animal Care and Use Committee (Application number 06340722-1).

Joint Public Review: https://doi.org/10.7554/eLife.96496.3.sa1
Author response https://doi.org/10.7554/eLife.96496.3.sa2

# Additional files

## Supplementary files

- Supplementary file 1. DNA and protein sequence for the PACLight1$_{P78A}$ sensor.
- MDAR checklist

## Data availability

The DNA and protein sequence of the sensor developed herein have been deposited on NCBI (accession number: OQ366523) and are available in Supplementary File 1. Viral DNA plasmids have been deposited both on Addgene (Addgene: Plasmid #197864) and on the UZH Viral Vector Facility (https://vvf.ethz.ch/; plasmids: v1030, v1031). Viral vectors can either be obtained from the Patriarchi laboratory or the UZH Viral Vector Facility. Data generated or analyzed during this study are available at https://doi.org/10.5281/zenodo.12699662. Source data are provided for Figures 1, 2, 3, 4, 5.

The following datasets were generated:

| Author(s) | Year | Dataset title | Dataset URL | Database and Identifier |
|---|---|---|---|---|
| Cola RB, Tommaso P | 2024 | Synthetic construct sensor PACLight1 gene, complete cds | https://www.ncbi.nlm.nih.gov/nuccore/OQ366523 | NCBI GenBank, OQ366523 |
| Patriarchi T, Cola RB, Niethammer SN, Rajamannar P, Gresch A | 2024 | Probing PAC1 receptor activation across species with an engineered sensor | https://doi.org/10.5281/zenodo.12699662 | Zenodo, 10.5281/zenodo.12699662 |

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
