## [Editor Report · eLife assessment]

This **fundamental** paper reports a new biosensor to study G protein-coupled receptor activation by the pituitary adenylyl cyclase-activating polypeptide (PACAP) in cell culture, ex vivo (mouse brain slices), and in vivo (zebrafish, mouse). **Convincing** data are presented that show the new sensor works with high affinity in vitro, while requiring very high (non-physiological) concentrations of exogenous PACAP when applied to intact tissues. The sensor has not yet been used to detect endogenously released PACAP, raising questions about whether the sensor can be used for its intended purpose. While further work must be pursued to achieve broad in vivo applications under physiological conditions, the new tool will be of interest to cell biologists, especially those studying the large and significant GPCR family.

---

## [Referee Report · Joint Public Review]

The manuscript "Engineering of PAClight1P78A: A High-Performance Class-B1 GPCR-Based Sensor for PACAP1-38" by Cola et al. presents the development of a novel genetically encoded sensor, PAClight1P78A, based on the human PAC1 receptor. The authors provide a thorough in vitro and in vivo characterization of this sensor, demonstrating its potential utility across various applications in life sciences, including drug development and basic research.

The main criticism of this manuscript after initial review is that the PACLight1 sensor has not been shown to detect the release of endogenous PACAP, whether in culture, in vivo, or ex vivo. The authors appear to be cognizant of this significant limitation (for a PACAP sensor) but no significant changes to address this limitation are provided in the revision.

While the sensor that is described here is new and the experimental results support the conclusions, the sensor reported here is not suited for the detection of endogenous PACAP release in vivo. In some respects, this manuscript could be seen as a stepping stone for further development either by the authors or other groups. Indeed, in many cases initial versions of genetically encoded sensors undergo substantial development post-publication, as exemplified by the evolution of GCaMP. However, the situation with the PAClight sensor reported here requires a different approach. Unlike GCaMP, which was one of the first genetically encoded calcium indicators, PAClight is another variant in a series of GPCR-fluorophore conjugates, following methodologies similar to those developed in the Lin Tian lab and the multiple GRAB-based sensors from Yulong Li's lab. These sensors have already demonstrated in vivo applicability, setting a standard that PAClight must meet or exceed to confirm its value and novelty.

Given that the title of the manuscript, "Probing PAC1 receptor activation across species with an engineered sensor," implies broader applicability, it potentially misleads readers about the sensor's utility in vivo, where "in vivo" should be understood as referring to the detection of endogenous PACAP release.

To align the manuscript with the expectations set by its title, it is crucial that the authors either provide substantial in vivo validation (ability to detect endogenous release of PACAP) or revise the title and the text to clarify that the sensor is primarily intended to detect exogenously applied PACAP. This clarification will ensure that the manuscript accurately reflects the sensor's current capabilities and scope of use.

---

## [Author Response]

The following is the authors’ response to the original reviews.

**Public Reviews:**

**Reviewer #1 (Public Review):**
Summary:The manuscript "Engineering of PAClight1P78A: A High-Performance Class-B1 GPCR-Based Sensor for PACAP1-38" by Cola et al. presents the development of a novel genetically encoded sensor, PAClight1P78A, based on the human PAC1 receptor. The authors provide a thorough in vitro and in vivo characterization of this sensor, demonstrating its potential utility across various applications in life sciences, including drug development and basic research.The diverse methods to validate PAClight1P78A demonstrate a comprehensive approach to sensor engineering by combining biochemical characterization with in vivo studies in rodent brains and zebrafish. This establishes the sensor's biophysical properties (e.g., sensitivity, specificity, kinetics, and spectral properties) and demonstrates its functionality in physiologically relevant settings. Importantly, the inclusion of control sensors and the testing of potential intracellular downstream effects such as G-protein activation underscore a careful consideration of specificity and biological impact.Strengths:The fundamental development of PAClight1P78A addresses a significant gap in sensors for Class-B1 GPCRs. The iterative design process -starting from PAClight0.1 to the final PAClight1P78A variant - demonstrates compelling optimization. The innovative engineering results in a sensor with a high apparent dynamic range and excellent ligand selectivity, representing a significant advancement in the field. The rigorous in vitro characterization, including dynamic range, ligand specificity, and activation kinetics, provides a critical understanding of the sensor's utility. Including in vivo experiments in mice and zebrafish larvae demonstrates the sensor's applicability in complex biological systems.Weaknesses:The manuscript shows that the sensor fundamentally works in vivo, albeit in a limited capacity. The titration curves show sensitivity in the nmol range at which endogenous detection might be possible. However, perhaps the sensor is not sensitive enough or there are not any known robust paradigms for PACAP release. A more detailed discussion of the sensors's limitations, particularly regarding in vivo applications and the potential for detecting endogenous PACAP release, would be helpful.

We thank the reviewer for carefully analyzing our in vivo data and highlighting the limitation of our results regarding the sensor’s applicability in detecting endogenous PACAP. We added several sections conversing future possibilities for optimization in the discussion (see paragraphs 2-4). We agree that a more specific discussion of the limitations of our study is an important addition to help design future experiments.

There are several experiments with an n=1 and other low single-digit numbers. I assume that refers to biological replicates such as mice or culture wells, but it is not well defined. n=1 in experimental contexts, particularly in Figure 1, raises significant concerns about the exact dynamic range of the sensor, data reproducibility, and the robustness of conclusions drawn from these experiments. Also, ROI for cell cultures, like in Figure 1, is not well defined. The methods mentioned ROIs were manually selected, which appears very selective, and the values in Figure 1c become unnecessarily questionable. The lack of definition for "ROI" is confusing. Do ROIs refer to cells, specific locations on the cell membrane, or groups of cells? It would be best if the authors could use unbiased methods for image analysis that include the majority of responsive areas or an explanation of why certain ROIs are included or excluded.

We thank the reviewer for the helpful suggestions. We have increased the number of replicates to n=3 for both HEK293T and neuron data depicted in Fig.1c. Furthermore, we have added Fig.1c’ containing the quantification of the maximum responses obtained in the dataset shown in Fig.1c also depicting the single values for each replicate. To clarify the definition of an ROI in our manuscript, we have detailed the process of ROI selection in the Methods section “Cell culture, imaging and quantification section”. Additionally, we also increased mouse numbers for in vivo PACAP infusions in mice (see Figure 4g).

**Reviewer #2 (Public Review):**
Summary:The PAClight1 sensor was developed using an approach successful for the development of other fluorescence-based GPCR sensors, which is the complete replacement of the third intracellular loop of the receptor with a circularly-permuted green fluorescent protein. When expressed in HEK cells, this sensor showed good expression and a weak but measurable response to the extracellular presence of PACAP1-38 aF/Fo of 43%. Additional mutation near the site of insertion of the linearized GPF, at the C-terminus of the receptor, and within the second intracellular loop produced a final optimized sensor with F/Fo of >1000%. Finally, screening of mutational libraries that also included alterations in the extracellular ligand-binding domain of the receptor yielded a molecule, PAClight1P78A, that exhibited a high ligand-dependent fluorescence response combined with a high differential sensitivity to PACAP (EC50 30 nM based on cytometric sorting of stably transfected HEK293 cells) compared to its congener VIP, (with which PACAP shares two highly related receptors, VPAC1 and VPAC2) as well as several unrelated neuropeptides, and significantly slowed activation kinetics by PACAP in the presence of a 10-fold molar excess of the PAC1 antagonist PACAP6-38. A structurally highly similar control construct, PAClight1P78Actl, showed correspondingly similar basal expression in HEK293 cells, but no PACAP-dependent enhancement in fluorescent properties.PAClight1P78A was expressed in neurons of the mouse cortex via AAV9.hSyn-mediated gene transduction. Slices taken from PAClight1P78A-transfected cortex, but not slices taken from PAClight1P78Actl-transfected cortex exhibited prompt and persistent elevation of F/Fo after 2 minutes of perfusion with PACAP1-38 which persisted for up to 14 minutes and was statistically significant after perfusion with 3000, but not 300 or 30 nM, of peptide. Likewise, microinfusion of 200 nL of 300 uM PACAP1-38 into the cortex of optical fiber-implanted freely moving mice elicited a F/Fo (%) of greater than 15, and significantly higher than that elicited by application of similar concentrations of VIP, CRF, or enkephalin, or vehicle alone. In vivo experiments were carried out in zebrafish larvae by the introduction of PAClight1P78A into single-cell stage *Danio rerio* embryos using a Tol2 transposase-based plasmid with a UAS promoter via injection (of plasmid and transposase mRNA), and sorting of post-fertilization embryos using a marker for transgenesis carried in the UAS :PAClight1P78A construct. Expression of PAClight1P78A was directed to cells in the olfactory bulb which express the fish paralog of the human PAC1 receptor by using the Tg(GnRH3:gal4ff) line, and fluorescent signals were elicited by intracerebroventricular administration of PACAP1-38 at a single concentration (1 mM), which were specific to PACAP and to the presence of PAClight1P78A per se, as controlled by parallel experiments in which PAClight1P78Actl instead of PAClight1P78A was contained in the transgenic plasmid.Major strengths and weaknesses of the methods and resultsThe report represents a rigorous demonstration of the elicitation of fluorescent signals upon pharmacological exposure to PACAP in nervous system tissue expressing PAClight1P78A in both mammals (mice) and fish (zebrafish larvae). Figure 4d shows a change in GFP fluorescence activation by PACAP occurring several seconds after the cessation of PACAP perfusion over a two-minute period, and its persistence for several minutes following. One wonders if one is apprehending the graphical presentation of the data incorrectly, or if the activation of fluorescence efficiency by ligand presentation is irreversible in this context, in which case the utility of the probe as a real-time indicator, in vivo, of released peptide might be diminished.

We thank the reviewer for their careful consideration of our manuscript and agree that the activation of PAClight persisting for several minutes at micromolar concentrations could be a potential limitation for in vivo applications. We added a possible explanation for the persisting sensor activation in response to artificial application of PACAP38 in paragraph 3 of the discussion. We agree that this addition eases the interpretation of PAClight signals detected in vivo.

Appraisal of achievement of aims, and data support of conclusions:Small cavils with controls are omitted for clarity; the larger issue of appraisal of results based on the scope of the designed experiments is discussed in the section below. An interesting question related to the time dependence of the PACAP-elicited activation of PAClight1P87A is its onset and reversibility, and additional data related to this would be welcome.

We agree that the reversibility of the sensor’s fluorescence is indeed an important feature especially for detecting endogenous PACAP release. Our data indicate that the sensor’s fluorescence is reversible when detecting small to medium doses of PACAP38 (see Figure 4d – Application of 30-300nM) that are presumably closer to physiological concentrations than the non-reversible concentration of 3000nM. Please, see also our new discussion on peptide concentrations in paragraph 4 of our discussion. For future experiments, it is indeed advisable to adjust the interval of repeated applications to the decay of the response at the respective concentration. Considering, the long-lasting downstream effects of endogenous signaling, longer intervals between ligand applications are generally preferred to match more closely the physiological range in which endogenous PAC1 is most likely affective.

Discussion of the impact of the work, and utility of the methods and data:Increasingly, neurotransmitter function may be observed in vivo, rather than by inferring in vivo function from in vitro, in cellular, or ex vivo experimentation. This very valuable report discloses the invention of a genetically encoded sensor for the class B1 GPCR PAC1. PAC1 is the major receptor for the neuropeptide PACAP, which in turn is a major neurotransmitter involved in brain response to psychogenic stress, or threat, in vertebrates as diverse as mammals and fishes. If this sensor possesses the sensitivity to detect endogenously released PACAP in vivo it will indeed be an impactful tool for understanding PACAP neurotransmission (and indeed PACAP action in general, in immune and endocrine compartments as well) in future experiments.However, the sensor has not yet been used to detect endogenously released PACAP. Until this has been done, one cannot answer the question as to whether the levels of exogenously perfused/administered PACAP used here merely to calibrate the sensor's sensitivity are indeed unphysiologically high. If endogenous PACAP levels don't get that high, then the sensor will not be useful for its intended purpose. The authors should address this issue and allude to what kind of experiments would need to be done in order to detect endogenous PACAP release in living tissue in intact animals. The authors could comment upon the success of other GPCR sensors that have been used to observe endogenous ligand release, and where along the pathway to becoming a truly useful reagent this particular sensor is.

We thank the reviewer for highlighting the lack in clarity that the scope of this paper was not intended to cover the detection of endogenous PACAP release. We therefore expanded our discussion to encompass the intended purpose of detecting artificially infused or applied PAC1 agonists, such as conducting fundamental tests of drug specificity and developing new pharmacological ligands to selectively target PAC1. This includes a more detailed discussion of our in vivo findings and a clearer phrasing that stresses the potential application for applied drugs and not endogenous PACAP (see last paragraph in the discussion).

We also agree that little is known about endogenous concentrations of PACAP in the brain. However, we have supplemented our discussion with several references estimating lower concentrations of PACAP and other peptides in vivo, suggesting average PACAP levels below the detection threshold of the sensor. Importantly, within certain brain regions and in closer proximity to release sites, significantly higher concentrations might be reached. Additionally, our data indicate that the concentrations observed under our current conditions do not saturate the sensor in vivo.

We therefore acknowledge the reviewer’s comment on the sensor’s potential limitations under our current experimental conditions. Hence, we expanded our discussion and suggest the use of higher resolution imaging to potentially reveal loci of high PACAP concentrations, which should be validated by future studies (see also our added discussion in paragraph 4).

**Reviewer #3 (Public Review):**
Summary:The manuscript introduces PAClight1P78A, a novel genetically encoded sensor designed to facilitate the study of class-B1 G protein-coupled receptors (GPCRs), focusing on the human PAC1 receptor. Addressing the significant challenge of investigating these clinically relevant drug targets, the sensor demonstrates a high dynamic range, excellent ligand selectivity, and rapid activation kinetics. It is validated across a variety of experimental contexts including in vitro, ex vivo, and in vivo models in mice and zebrafish, showcasing its utility for high-throughput screening, basic research, and drug development efforts related to GPCR dynamics and pharmacology.Strengths:The innovative design of PAClight1P78A successfully bridges a crucial gap in GPCR research by enabling realtime monitoring of receptor activation with high specificity and sensitivity. The extensive validation across multiple models emphasizes the sensor's reliability and versatility, promising significant contributions to both the scientific understanding of GPCR mechanisms and the development of novel therapeutics. Furthermore, by providing the research community with detailed methodologies and access to the necessary viral vectors and plasmids, the authors ensure the sensor's broad applicability and ease of adoption for a wide range of studies focused on GPCR biology and drug targeting.WeaknessesTo further strengthen the manuscript and validate the efficacy of PAClight1P78A as a selective PACAP sensor, it is crucial to demonstrate the sensor's ability to detect endogenous PACAP release in vivo under physiological conditions. While the current data from artificial PACAP application in mouse brain slices and microinfusion in behaving mice provide foundational insights into the sensor's functionality, these approaches predominantly simulate conditions with potentially higher concentrations of PACAP than naturally occurring levels.

We thank the reviewer for their valuable comments and agree that the use of PAClight for detecting endogenous PACAP will be of big interest for the scientific community and should be a goal for future research. Considering the time, equipment and additional animal licenses necessary, we are convinced that these questions would go beyond the scope of the current paper and might rather be addressed in a follow-up publication. We therefore rephrased the discussion and added more details to clarify further the intended purpose of the current study. Additionally, we added a paragraph in the discussion suggesting experiments needed to validate PAClight for putative future in vivo applications.

Although the sensor's specificity for the PAC1 receptor and its primary ligand is a pivotal achievement, exploring its potential application to other GPCRs within the class-B1 family or broader categories could enhance the manuscript's impact, suggesting ways to adapt this technology for a wider array of receptor studies. Additionally, while the sensor's performance is convincingly demonstrated in short-term experiments, insights into its long-term stability and reusability in more prolonged or repeated measures scenarios would be valuable for researchers interested in chronic studies or longitudinal behavioral analyses. Addressing these aspects could broaden the understanding of the sensor's practical utility over extended research timelines.

We extend our gratitude to the reviewer for diligently assessing our results.

Indeed, the very high level of sensitivity that we could achieve in PAClight leads us to think that potentially a grafting-based approach, such as the one we’ve recently described for class-A GPCR-based sensors (PMID: 37474807) could also work for the direct generation of multiple class-B1 sensors based on the optimized fluorescent protein module present in PAClight. Unfortunately, considering the amount of work that testing this hypothesis would entail, we are not able to perform these experiments in the context of this revision, and would rather pursue them as a future project. Nevertheless, we have expanded the discussion of the manuscript with a paragraph with these considerations.

While we lack comprehensive data on the long-term stability of the sensor, our preliminary findings from photometry recordings optimization indicate consistent baseline expression of PAClight and PACLight ctrl over several weeks. Conducting experiments to systematically assess stability would require several months, which is currently impractical due to limitations in tools and licenses for repeated in vivo infusions. Hence, we intend to include these experiments in potential follow-up studies.

Furthermore, the current in vivo experiments involving microinfusion of PACAP near sensor-expressing areas in behaving mice are based on a relatively small sample size (n=2), which might limit the generalizability of the findings. Increasing the number of subjects in these experimental groups would enhance the statistical power of the results and provide a more robust assessment of the sensor's in vivo functionality. Expanding the sample size will not only validate the findings but also address potential variability within the population, thereby reinforcing the conclusions drawn from these crucial experiments.

We agree with the reviewer that a sample size of N=2 is not sufficient for in vivo recordings. We therefore increased the sample size and now present recordings with 5 PAClight1P78A and 4 PACLight-control mice. Of note, the new data validate our previous findings and conclusions and give a better idea of the variability in vivo that we now discuss in much more detail in the discussion (see paragraph 2).

**Recommendations for the Authors:**

**Reviewer #1 (Recommendations For The Authors):**
The lower potency of maxadilan activation might reflect broader implications for ligand-receptor dynamics. Perhaps the authors could discuss the maxadilan binding from a structural perspective, including AlphaFold models. Also, discussing how these findings might influence sensor application in diverse biological contexts would be insightful. Clear definitions and consistent use of these terms are crucial for ensuring that readers understand the methods and results.

We would like to thank the reviewer for the comments. As part of this work, we did not obtain a dose-response curve for maxadilan peptide, and only reported the maximal response of the sensor to a high concentration of the peptide (10 µM). Thus, our findings would rather inform us on the maximal efficacy of the peptide, as opposed to its potency towards the PAC1R. Furthermore, we would like to point out that due to the lack of structural details for any GPCR-based sensor published to date, we cannot make any molecularly accurate conclusion regarding the precise reasons why a different ligand (in this case the sandfly maxadilan) induces a lower maximal efficacy of the response compared to the endogenous cognate ligand of the receptor. We do not believe that AlphaFold models can accurately replace structural information in this regard, especially given the consideration that the aminoacid linker regions between the GPCR and the fluorescent protein, which are a critical determinant of allosteric chromophore modulation by ligand-induced conformational changes, typically obtain the lowest confidence score in all AlphaFold predicted structural models of GPCR-based sensors. Finally, we would like to refer the reviewer to a very nice recent publication (PMID: 32047270) which resolved the structures of each of these peptides bound to the PAC1 receptor-Gs protein complex, which provides accurate molecular details on the different modalities of receptor binding and activation by PACAP138 versus maxadilan.

**Reviewer #2 (Recommendations For The Authors):**
The authors are congratulated on the meticulous achievement of their aim, i.e. a fluorescence-based sensor for the detection of PACAP with in vivo utility. Whether or not this sensor will have the requisite sensitivity to detect the release of endogenous PACAP within various regions of the nervous system, in response to specific environmental stimuli or changes in brain or physiological state, remains to be determined.

We thank the reviewer for the very positive evaluation of our manuscript and for the suggested additions that will improve the strength of our arguments.

We agree that the in vivo detection of endogenous PACAP will be an important objective for future studies. Due to time, resource and animal license constraints, we are not able to address this objective in our current study, but we now detail possible future experiments in the discussion section. Please see also our answer to the suggested discussion points previously.

**Reviewer #3 (Recommendations For The Authors):**
To comprehensively assess the sensor's sensitivity and specificity to endogenous PACAP, I recommend conducting additional in vivo experiments where PAClight1P78A is expressed in neurons that endogenously express the Pac1r receptor (using Adcyap1r1-Cre mouse line). These experiments should involve applying sensory or emotional stimuli known to evoke PACAP release or activating upstream PACAP-expressing neurons. Such studies would offer valuable data on the sensor's performance under natural physiological conditions and its potential utility for exploring PACAP's roles in vivo.

We express our gratitude to the reviewer for providing detailed methodological approaches to examine endogenous PACAP release. These suggestions will prove invaluable for future investigations and are important additions to a follow-up publication. As mentioned earlier, we have incorporated some of these approaches into our discussion. Additionally, we have underscored the existing limitations in detecting endogenous PACAP in vivo and emphasized the relevance of PAClight for drug development purposes.